# DNA Based and Stimuli-Responsive Smart Nanocarrier for Diagnosis and Treatment of Cancer: Applications and Challenges

**DOI:** 10.3390/cancers13143396

**Published:** 2021-07-06

**Authors:** Fakhara Sabir, Mahira Zeeshan, Ushna Laraib, Mahmood Barani, Abbas Rahdar, Magali Cucchiarini, Sadanand Pandey

**Affiliations:** 1Faculty of Pharmacy, Institute of Pharmaceutical Technology and Regulatory Affairs, University of Szeged, Eötvös u. 6, H-6720 Szeged, Hungary; fakhra.sabir@gmail.com; 2Department of Pharmacy, Faculty of Biological Sciences, Quaid-i-Azam University, Islamabad 45320, Pakistan; mzeeshan@bs.qau.edu.pk; 3Department of Pharmacy, College of Pharmacy, University of Sargodha, Sargodha 40100, Pakistan; ushnalaraib@yahoo.com; 4Medical Mycology and Bacteriology Research Center, Kerman University of Medical Sciences, Kerman 76169-13555, Iran; mahmoodbarani7@gmail.com; 5Department of Physics, Faculty of Science, University of Zabol, Zabol 98615-538, Iran; a.rahdar@uoz.ac.ir; 6Center of Experimental Orthopaedics, Saarland University Medical Center, 66421 Homburg, Germany; 7Department of Chemistry, College of Natural Science, Yeungnam University, 280 Daehak-Ro, Gyeongsan 38541, Korea

**Keywords:** nanotechnology, DNA nanostructures, stimuli-responsive, smart nanocarriers, diagnosis, treatment, cancer therapy, drug delivery, theranostics

## Abstract

**Simple Summary:**

DNA based and stimuli-responsive smart nano-carrier system can efficiently deliver encapsulated cargos in response to various driving thresholds. The conditional release of encapsulated carriers via modifications in the dynamic structure of DNA in response to tumor microenvironment has led to the specific delivery of active agents to the tumors. Therefore, specific drug delivery enhances the therapeutic efficacy and decreases the side effects on the neighboring healthy cells. Despite the great potential for pharmacological applications of the DNA-assisted and pH-sensitive smart nano-carriers, some challenges should be addressed before clinical transition. Since the therapeutic efficacies of DNA based smart nano-carriers have only been investigated in a proof-of-concept manner, the detailed in vivo parameters of the nano-carriers must be investigated and elucidated, including colloidal stability, circulating half-life, immunogenicity, dose-dependent cytotoxicity, pharmacokinetics, and clearance mechanisms.

**Abstract:**

The rapid development of multidrug co-delivery and nano-medicines has made spontaneous progress in tumor treatment and diagnosis. DNA is a unique biological molecule that can be tailored and molded into various nanostructures. The addition of ligands or stimuli-responsive elements enables DNA nanostructures to mediate highly targeted drug delivery to the cancer cells. Smart DNA nanostructures, owing to their various shapes, sizes, geometry, sequences, and characteristics, have various modes of cellular internalization and final disposition. On the other hand, functionalized DNA nanocarriers have specific receptor-mediated uptake, and most of these ligand anchored nanostructures able to escape lysosomal degradation. DNA-based and stimuli responsive nano-carrier systems are the latest advancement in cancer targeting. The data exploration from various studies demonstrated that the DNA nanostructure and stimuli responsive drug delivery systems are perfect tools to overcome the problems existing in the cancer treatment including toxicity and compromised drug efficacy. In this light, the review summarized the insights about various types of DNA nanostructures and stimuli responsive nanocarrier systems applications for diagnosis and treatment of cancer.

## 1. Introduction

According to Globacon 2018 [1,2], cancer, the second leading cause of death, claimed the lives of approximately 9.6 million people worldwide. It accounts for 13 percent of all the deaths every year and this mortality rate is anticipated to rise to 13.1 million by 2030 as per WHO statistics [3]. According to the reports of the World Health Organization, every 1 in 6 females and 1 in 5 males develop cancer at some time in their life whereas every 1 in 11 females and 1 in 8 males lose their lives to this malignant disease [4].

Cancer is the uncontrolled proliferation of cells that progresses through multistep carcinogenesis. An obscure disease demands complex approaches for treatment [5]. Conventional approaches to treat cancer include chemotherapy, surgery, irradiation, immunotherapy, or a combination of these procedures [6]. All these procedures yield inadequate therapeutic efficiency owing to the unwanted side effects. Traditional chemotherapeutic agents may not work for all types of cancer [7]. These drugs being poorly soluble have limited targeting and delivery to malignant tumors. Additionally, these therapeutic agents have non-specific nature leading to the undesirable killing of the normal healthy tissues along with cancerous tissues. At higher doses, these agents lead to life-threatening toxicity [8,9]. One major drawback of these anti-cancer drugs is the development of multiple drug resistance (MDR) which fails chemotherapy [10]. Concurrent administration of chemotherapy with irradiation, i.e., chemo-radiotherapy is another standard method in cancer treatment. Nevertheless, the combination therapy causes a significant increase in toxicity, which is greater than the individual treatment procedures [11].

Another major challenge in the efficacy of cancer treatments is the early detection and diagnosis of malignant tissues. Current imaging techniques (e.g., ultrasound, MRI, CT, and X-ray) only detect malignancy when there is an apparent change in tissue. However, by that time the cancerous cells not only have proliferated but metastasized as well [12].

The current statistics, mortality rate, and limitations of conventional approaches to treat cancer cells for more advanced technology. Over the past decade, researchers have shown great interest in nanotechnology, which is exhibiting significantly improved patient outcomes. Nanoparticle based drug delivery systems are being extensively implemented in conventional approaches and combinational therapies of cancer treatment. These nanocarriers have shown improved efficacy, selectivity, stability, bio distribution, biocompatibility, and pharmacokinetics of anti-cancer drugs while overcoming the issues of multiple drug resistance and toxicity [13,14]. A variety of payloads, e.g., imaging agents, antibodies, photosensitizers, nucleic acids, and anti-cancer drugs can be loaded into these nanocarriers making them a versatile platform. In addition, nanocarrierscan be developed into cancer ‘theranostics’ by combining both therapeutic and diagnostics agents [15,16,17].

Nanocarriers are carrier modules for substances or drugs and having a size range of 500 nm that can prompt changes in properties of drugs and their bioactivation [18]. However, the conventional nanocarrier drug systems are not only unable to release the drug at the right concentration but are also prone to prematurely releasing its contents, damaging the healthy tissues along with decreased therapeutic efficacy. The drawbacks of conventional nanoparticle based drug delivery system led scientists to develop smart or intelligent drug delivery systems that can release their contents at the targeted site in response to a particular trigger. Stimuli based nanocarriers have modified structures that are sensitive to internal (temperature, pH, redox, enzymes reactions) or external (ultrasound, electric field, light, magnetic field) environment [19]. Nanocarriers can be organic, inorganic, or hybrid and follow either an active mechanism or a passive mechanism for tumor targeting. Choice of material for synthesizing nanocarriers is mostly based on: (1) therapeutic outcome desired or diagnosis; (2) safety profile of the material; (3) route of administration; (4) type of payload [18,20].

Further development in this area has established high gene loading potential of polymeric nanocarriers. Gene therapy is another alternative but effective method for treating cancer. Smart nanocarriers as non-viral gene delivery drug delivery systems do not only overcome the safety issues associated with conventional non-viral vectors but are also able to deliver the genetic material to target sites for either expressing therapeutic proteins or blocking gene expression [21]. In response to a single, dual, or multiple stimuli, these smart carriers exhibit structural changes. For example, light responsive nanocarriers at a particular wavelength of light can control gene release. Similarly, encapsulation of iron particles or exposing a tumor site to ultrasound controls the release in magnetic and ultrasound responsive gene delivery systems, respectively [22]. Owing to the highly specific and tunable nature of DNA, researchers were able to develop smart DNA-assisted nanocarriers tailored with switchable DNA framework, which acts as the driving force for the release of therapeutic drugs in response to a stimulus. This conditional release leads to an increased therapeutic efficacy with minimal damage to the healthy neighboring cell [23]. The review aims to provide insights into various types of DNA-based smart nanocarrier systems applicable in the diagnosis and treatment of cancer and to briefly summarize various synthesis strategies and mechanisms of internalization for these systems.

## 2. Classification and Applications of Smart Nanocarrier System in Cancer Targeting

DNA is a novel and smart biomaterials that can be implied to synthesize the various types of nanocarriers system based on its key GC/AT complementary base pairing. The data from numerous studies revealed that the DNA nanostructure is an effective tool for addressing major issues in cancer care, such as toxicity and drug efficacy. Therefore, some significant improvements have been made in recent years [24]. One of the advancements to these intelligent nanocarrier systems is multidrug co-delivery, which increases the targetability with the help of various ligands and adaptations of active target strategies. Different methods are adopted for the preparation of DNA based nanocarrier systems. Basically, these nanocarrier systems consist of functional DNA sequences, biomloecules, that are bound using physical, chemical, or biological engineering tools. Back into the history regarding the evolution of DNA based nanocarrier system, first static four-arm structure of DNA was designed by Nadrain Seeman in 1983, consisting of four strands of DNA. Each strand has a different base sequence to make a junction point at specific loci [25]. These static DNA joints are basic blocks to design a stable and rigid DNA nanostructures. With further advancement in this field more arms including three, five, six, eight, twelve were generated for the production of various DNA nanostructures. Structural DNA nanotechnology has become significantly important in the field of nanoscience since the 1980s [26]. The various dynamic and static data DNA devices with various dimensions and structures have been introduced and developed. The pure DNA consisted of nanostructure have been divided into various types as shown in schematic representation (Figure 1) of different classification of DNA based nanosystem [27].

The DNA polyhedron nanosystems have been designed from tetrahedron and DNA octahedron to DNA icosahedron that served as simple carriers in anti-cancer drug delivery. Tuberfield and his colleagues developed classic DNA tetrahedron for the first time. Thereafter, it has been used as an efficient cargo for anti-cancer drugs including photosensitizer, DOX, siRNA, and other drugs concurrently. The anti-neoplastic drugs (Dox, doxorubicin) encapsulated into DNA tetrahedron can kill the circulating tumor cells (CTC) [23,28]. Furthermore, the light will cause the photosensitizer marked on the DNA tetrahedron, resulting in enhanced cytotoxic effects. There are several abilities of DNA nanodevices to increase the endocytotic uptake of anti-neoplastic agents and also increased the drug loading capacity with greater efficacy. Most of the present literature study data emphasize the progress in modification to increase the drug ability and to decrease its adverse effects [29]. One research group created a DNA tetrahedron to encapsulate DOX with available conjugation sites for attaching cetuximab antibodies that target the epidermal growth factor receptor specifically. The findings of the following study showed that this nanosystem have greater targeting ability and better killing efficacy of malignant cells. Chen et al. developed biotins conjugated to DNA tetrahedron (ruthenium polypyridyl complexes). The DNA cage also increases its specific cellular uptake along with drug cytotoxicity and retention against HepG2 cells [30,31].

Lo’s group has produced a DNA nanocage for the first time for mitochondrial delivery of DOX by conjugating lipids. In contrast to DOX localization in lysosomes, DOX retention in mitochondria causes major cytotoxicity and cellular apoptosis in MCF-7 cells, according to the findings. However, with the introduction of stimuli responsive DNA tetrahedrons and switchable DNA nanosuitcases, more stimuli responsive DNA polyhedron drug delivery strategies will be established and used in advanced nanotechnology cancer treatment [32,33].

Aside from hybridization, catalytic hairpin conjugation may generate DNA nanoribbons. Rigid and programmable DNA tiles have previously been used to cause significant one-dimensional (1D) nanoribbons, nanotubes, two-dimensional (2D) arrays, and even three-dimensional (3D) crystals [34]. By use of different technologies number of researchers design different nanoribbons to deliver the siRNA, DOX, photosensitizer, and so on [35]. Weizmann et al. developed DNA nanoribbon by a modified DNA origami strategy. Furthermore, various studies proved that the DNA nanoribbons was an efficient siRNA delivery cargo in human cells cancer. The functionalized DNA nanoribbon structures and devices show extraordinary performance in cancer diagnosis and treatment because of their small sizes, morphology, and greater biocompatibility. Several research groups collaborated to develop various types of DNA nanoribbons, for example, Liang et al. developed DNA nanoribbon with two compartments, one was loaded with -GC- base pairs for DOX delivery. Another component was the AS1411 aptamer, which is a DNA aptamer. The following system helped to increase the tolerability of human breast cancer cells to the DOX with inhibition of tumor cell proliferation. Self-assembled DNA nanocentipede was developed by Roh et al. to deliver multivalent aptamers as to functionalize in cancer targeting [36,37]. Chu’s and his colleagues developed an aptamer probe to target the cancer cells via structure switching. Hybridization chain reaction (HCR) accumulated higher encapsulated prodrugs from a drug labeled probe and induced their conversion and uptake into cisplatin in cells for selective tumor targeting using this strategy [38]. Another type of DNA assembly nanosystems designed by a group of researchers. They classified these materials into two groups: DNA nanohydrogels and DNA dendrimers [39]. DNA dendrimers are basically hybridized layer by layer self-assembled functional branched DNA [40]. DNA nanohydrogels, on the other hand, are made from functional building blocks by base-pairing hybridization or liquid crystallization and dense packaging. Since they can be configured into and provide further docking sites to encapsulate drugs or other functional elements, these DNA nanosystems are denser. Yang et al. developed DNA dendrimers and encapsulate DOX. Other researcher group developed nanohydrogels from hybridization of different building blocks to synergistic cancer therapy with Dox [41,42]. Different researchers have applied different methods for the development of DNA nano-hydrogels for targeting DOX delivery by using building blocks and liquid crystallization without base-pairing hybridization. The DNA nanohydrogel is comprised of three building blocks unit including functional moiety, DNAzyme, and aptamer. Each of these parts have different functionalization. These three parts self-assembled into nanohydrogels by hybridization between sticky ends [43].

DNA nanoflower system in comparison to self-assembled, form long DNA strands via rolling circle replication along with liquid crystallization and dense packaging. Despite the drawbacks of large nanostructures, the above type of nanostructure seems to have its own set of characteristics. To deliver anti-cancer drugs, this form of structure is very light in sequence design and its size can be tuned by varying the assembly time and template sequence. A group of researchers had develop series of nanoflowers to encapsulate the anti-cancer agents (CpG, DOX). Furthermore, the researchers modify the nanoflowers to encapsulate different types of agents for multigene therapy [44,45].

Since DNA origami is large and dense, it has a high ability to target tumors without the need for targeted modification. The first DNA origami design was complicated because it relied on the hybridization of a long ssDNA from the M13 phage genome with hundreds of short staple strands. However, further improvements in this design were implied by many researchers to simplify the method of its development by using RCP-amplified scaffold in replacement to ssDNA from M13 phage [46]. Likewise, with other DNA based nanostructures it is efficient for DOX, CpG, photosensitizer, etc. More advanced DNA origami structures include DNA rod/tube and DNA triangle to encapsulate a high load of drugs. Another study used DOX encapsulated DNA origami delivery systems that can induce remarkable cytotoxicity in cancer targeting. Bachelet et al. designed a hollow hexagonal barrel-shaped DNA origami as a wonderful logic gated nanorobot to handle the release of encapsulated molecules while identifying specific receptor proteins [47]. Following that, they build more complex nanorobots that can interact with one another and generate logical outputs to turn molecular payloads on or off [48].

DNA structure further classified into hybrid DNA nanostructured system that is subdivided into DNA-inorganic nanoparticle (non-stimuli responsive, light responsive, small molecule, DNA lipid hybrid, DNA polymer hybrid nanosystems, and small active substance responsive nanosystem) [49]. DNA-inorganic nanoparticles hybrid system including non-stimuli responsive systems have been designed for better cancer treatment. This system included both non-stimuli responsive and stimuli responsive nanocarrier systems, which are commonly constructed, based on the change DNA configuration [50]. Present literature mentioned that nanoflower inorganic nanoparticles have a spherical shape and increased the concentration of drug at the malignant site [50,51]. They developed AS1411/magnetic nanoparticles for targeted TMPyP4 delivery in this type of non-stimuli inorganic nanoparticle method. They also developed a Sgc8/MNP nanosystem and peptide/Au NPs for targeted DOX delivery [52,53]. Jiang and Zhang et al. engineered DNA nanoflower/polyhedron on nanoparticles for DOX delivery and photosensitizer co-delivery [54]. Ding et al. developed a triangle DNA origami-gold nanorod complex that showed distinguish increase in cellular uptake and enhanced photothermal effect of Au against tumor cells. Light responsive nanosystems used dsDNA to connect with inorganic nanoparticles. AuNPs are representative of light responsive nanosystem because AuNPs can convert light into heat to assist in the degradation of dsDNA and further release of drugs [55]. Huang’s group developed AS1411 aptamer conjugated dsDNA hybrid nanostructures for co-delivering of Dox and TMPyP4. By applying heat or light effect on Au-Ag nanorod drug can be accumulated in higher concentration in the nuclei to efficiently kill the cancer cells. In a study, mesoporous silica nanoparticles were developed to perform on-demand stimuli response of therapeutics. Single-stranded DNA was ligated to magnetic nanoparticles. Magnetic nanoparticles were then decorated with complementary DNA sequences. The uncapping and subsequent release of mesopore filled model drug was caused by DNA double stranded melting as a result of temperature increase [56]. DNA lipid hybrid nanosystems are another type of DNA assessed delivery system, where functional DNA can connect to lipid to form hybrid nanosystems for tumor targeting. DNA polymer hybrid nanosystems are called hybrids as they can self-assemble into spherical structures without any complex design structure. Additionally, they are very supportive for other active agents like paclitaxel in the hydrophobic parts. Another type of DNA nanocargo includes polymer hybrid nanosystem that have greater encapsulation efficiency and can protect the drug against premature degradation. This property of the polymer hybrid system further helped to design the stimuli responsive nanosystem [57]. Willners et al. developed poly-function core and multilayer shell based DNA polymer hybrid system for controlled release. Table 1 demonstrated the prerequisite of DNA nanostructures alongwith their surface characteristics for particular organ targeting. These specific DNA assemblies were designed to identify the specific stimuli like pH, light, ATP to modify their conformation for drug release [58].

DNA based nanosystems developed circular DNA nanotechnology for ligand functionalization (neuregulin-1/NRG-1) and its biological application [63]. A group of researchers developed DNA nanospindals (DNA-NS) to efficiently load daunorubicin (DR) and target the HER2/neu receptors on the plasma membrane of drug-resistant MCF-7 (breast cancer) cells. DR loading onto DNA-NS was confirmed by the UV-shift analysis. The MTT results showed reduced viability of the MCF-7 cells after treatment with DNA-NS. Further results of apoptosis/proliferation obtained via flow cytometry showed enhanced apoptosis up to (64%) after treating with DNA-NS. Hence, all the types of DNA nanostructures in cancer therapy showed stiffer, uniform and more biocompatible-targeted therapy [64]. Figure 2 shows types of DNA nanostructures.

## 3. DNA Assessed Stimuli Responsive Nanoparticle System for Cancer Targeting

### 3.1. Exogenous Stimuli Responsive Nanocarrier System for Diagnosis and Treatment of Cancer

The exogenous stimuli including magnetic, thermal, electronic, ultrasound light field can affect the internalization of nanocarriers inside the biological systems. The application of external stimuli has several advantages for targeting delivery to tumors: (I) the location and intensity of applied stimuli could be precisely controlled; (II) the stimuli can be added or removed based on available treatment requirements; (III) several different types of stimuli could be used for multifunction in cancer theranostics; (IV) the possibility to provide continuous or multi-times stimuli for drug therapy and delivery [65]. Table 2 shows different types of stimuli for gene and drug delivery for cancer targeting.

### 3.2. Ultrasound Responsive Nanocarriers

A high-intensity sound wave could affect nanocarriers for controlled release at malignant sites. For various applications, the ultrasound intensity may be changed. The ultrasonic intensity can be modified for different uses. At low frequencies, it could be used for imaging, and at higher frequencies, it could be used as a catalyst to release drugs from nanocarriers or increase the permeability of malignant cell membranes. There are several sizes of micro bubbles have been developed for ultrasound imaging and also commercialized as Albunex, Sonazoid, Optison, etc [63,74]. Microbubbles’ large size (1–10 µm), short half-life, and low stability restricted access to the vascular compartment in tumor tissues. As a result, several switchable microbubbles or nanocarriers for ultrasound imaging have been produced. The ultrasound sensitive nanocarriers include air, perfluorocarbons, N_2_, etc. or gases releases in biological environment.

Porphyrin microbubbles (1–10 µm) may be transformed into nanobubbles (5–500 nm) for tumor imaging using an ultrasound-responsive nanocarriers strategy [75]. Due to the collapse of microbubbles in response to low intensity ultrasound waves, phase-changeable polymeric nanoparticles could be produced for tumor imaging and doxorubicin release. The large size of ultrasound-sensitive nanoparticles may limit the penetration across the malignant site. Furthermore, the drug encapsulated ultrasound sensitive nanomaterials can be applied for the tumor application, theranostics, and image-guided therapy. One study group developed nanocarrier emulsion made up of perfluoropentane nanodroplet within the aqueous layer of a liposome, along with anticancer drug doxorubicin [76]. The liposomes encapsulated with DOX showed it release on insonation with low intensity ultrasound at 20-kHz, 1.0 MHz, and 3.0 MHz. More release occurs in vitro at 20 kHz than at greater frequencies. The results showed that this system promises to have more efficient therapy and tumor treatment to decrease the adverse effects of cardiotoxicity caused by Dox. In another research, liposomes were encapsulated with docetaxel and NH_4_HCO_3_ to generate CO_2_ bubbles in tumors for dual ligand-based targeted delivery and ultrasound imaging. One study claimed multimodal ultrasound imaging and molecular biosensors application of nanodroplets bubble vesicles by using genetically encoded nanostructure from microorganisms [77].

Gaspar et al. developed DOX and DNA micelleplexes for co-delivery via stimuli sensitive polymeric nanocarriers. The obtained results showed that minicircle DNA (mcDNA) encapsulated micelleplexes into in vitro tumor spheroid models with specific kinetic and show enhanced gene expression in comparison to other nanocarriers. Moreover, dual-loaded micelleplexes showed a significant uptake and cytotoxic effect in cancer. The findings revealed that triblock micelles are effective for in vivo delivery and have the potential to be used in DNA therapy. Gaspar et al. developed a gas penetrating stimuli sensitive hollow microspheres as a strategy to co-deliver Dox and minicircle DNA. The results demonstrate that microcarriers produced gas mediated Dox release and dual loaded particles achieved 5.2-fold greater cellular internalization in comparison to non-pegylated microspheres [78]. A stronger cytotoxic effect occurred from the increased cellular concentration. The enhanced transgene expression was obtained after nanoparticle-mcDNA co-delivery in the microspheres. The results showed that nanoparticle-microsphere systems to achieve efficient co-delivery of different drug-mcDNA combinations [79]. Figure 3 demonstrate the application of liposomes nanosystem to the cancer site. The stimuli used was ultrasound that releases the payloads with insonation at low intensity to the targeted cell.

### 3.3. Magnetic Field Triggered Therapy

Magnetic stimulation candidates include core shell-dependent nanoparticles coated with silica polymer or magnetoliposome. Magnetically coated nanoparticles may also be used to transport genetic information. When held under an oscillating magnetic field, magnetic nanocarriers can generate heat in close proximity. The structure of nanocarriers can be altered by heat. Attractive Magnetic nanoparticles (MNPs) with the ability to react to a magnetic field can be used in gene and drug delivery using magnetic targeting. Different malignant cells, such as brain, lung, breast, and prostate cancer, have been targeted with magnetic targeting. Similarly, a magnetic field may cause the targeted transmission to a specific location, and MNPs have been used to transfect DNA and RNA [65,80]. The drug delivery system based on MNPs not only delivers the drugs to a particular location but also regulates their release. Drugs can be attached to MNPs by conjugation on a heat sensitive linker or through p-p interaction and in some situations by co-embedding within thermally sensitive polymers. Under an alternating magnetic field, MNPs can produce heat that can improve the drug release due to the cracking of the polymer or linker [81]. The MNPs heat can generate pressure inside the porous NPs, triggering the drug release. Dobson et al. attributed it to the association of magnetic vectors with membranes and transmission of mechanical forces from the lateral movement of the magnetic field to cellular membranes [82]. The magnetic materials can be applied for tumor imaging via magnetic resonance imaging (MRI). Moreover, besides contrast agents’ plasmids, anti-bodies, photosensitizer can also be incorporated inside the magnetic sensitive nanoparticles to achieve multiple multimodal therapeutic effects. The alternating magnetic field sensitive hyperthermia can induce the release of drug from nanocarriers in diseased regions that is tumor or cancer cells [83]. The PEGylated MoS_2_/Fe_3_O_4_ nanocomposites made via two-step hydrothermal method have shown greater efficiency for tumor targeting. The two-step hydrothermal method demonstrated greater potential for tumor diagnosis by T2-weighted imaging and photoacoustic tomography. Moreover, it further allowed both T1 and T2 weighted MRI of tumors by doping Mn into core of Fe_3_O_4_@MoS_2_ multifunctional nanoflowers [84].

### 3.4. Thermo-Responsive Nanocarriers Applied for Diagnosis and Treatment of Cancer

Thermo-responsiveness can be defined as the ability of a substance or material to undergo drastic changes in at least one of its physicochemical properties upon variation in temperature [85]. Due to the phase transition behavior, tunable and versatile design, temperature responsive polymers have been extensively studied as smart drug delivery systems [86]. A temperature change can be easily controlled and implemented in vitro/in vivo with convenience. Temperature is also a unique stimulus than others as it can be utilized as an external as well as an internal stimulus.

Temperature acts as an external stimulus when heat is provided from outside of the body or by irradiation, electric field, magnetic field, etc. External heating can also result in the direct killing of cancer cells, as they are naturally susceptible to heat. Temperature is utilized as an internal stimulus when certain pathological conditions elevate the temperature of the specific site in the body. Due to the Warburg effect, tumors show a slight 2–3 degree elevated temperature (40–42 °C) than the normal tissues (37 °C). A change in temperature around the drug-carrying system leads to a sharp non-linear change in the temperature sensitive element of the carrier system resulting in drug release. Ideally, these nanocarriers should be able to maintain the drug load at normal body temperature and should only release the drug in an elevated temperature environment [87,88,89].

To date, many thermo-responsive nanocarriers have been successfully synthesized including liposomes, nanocomposites, nanogels, polymeric micelles, nanocapsules and vesicles. These nanocarriers are either developed with a material that changes their physicochemical properties upon variation in temperature or by incorporating a thermally unstable polymer [16]. For example, liposomes incorporated with NH_4_HCO_3_ generated CO_2_ from local hyperthermia of tumor resulting in swelling and collapsing of the system. This resulted in an efficient drug release [90].

Generally, temperature responsive materials or polymeric nanoparticles can be prepared from techniques like free radical polymerization followed by hydrolysis, phase separation, emulsion, foaming and graft copolymerization mediated by UV irradiation, etc. [91]. Recently advanced polymerization techniques are being used for developing and functionalizing new thermo-responsive polymers. Reversible deactivation radical polymerization (RDRP) techniques, which include atom transfer radical polymerization (ATRP), nitro-oxide mediated polymerization (NMP), and reversible addition-fragmentation chain transfer (RAFT) enables the development of complex macromolecular structures with low variance and high chain-end precision along with other wide range of functionalization options. Ring opening polymerization (ROP) technique allows the synthesis of well-defined polymers [85].

The fundamental principle of thermo-responsive polymers is based on critical solution temperature (CST). These polymers exhibit a change in their solubility in response to changes in temperature. CST is a temperature at which separation of polymer phase occurs. CST is further divided into lower critical solution temperature (LCST) or upper critical solution temperature (UCST) [92]. Controlled drug delivery systems can be achieved by controlling LCST or UCST which results in phase transition followed by either swelling or shrinking. Majority polymers are synthesized based on their LCST. The LCST transition is dependent on the nature of the polymer rather than the carrier state like micelles, hydrogels, etc. Below the LCST, the polymer exists in a monophasic and hydrophilic state. Above the LCST it exists in an insoluble, biphasic and hydrophobic state [24]. At this stage the polymer solution becomes cloudy and the effect is known as the ‘cloud point’. This effect is related to the concentration of the polymer and other constituents [93]. An increase in temperature above LCST disintegrates the network due to coil to globule transition. As a consequence, volume shrinkage occurs that forces the encapsulated contents to squeeze out and subsequently drug release. Such polymers are termed as negative thermosensitive polymers [91].

In case of UCST polymers, the increase in temperature above UCST increases the solubility of the polymer and subsequently swelling. However, only a little research has been conducted on these thermo-positive polymers. It should be noted that the changes in the volume are reversible and referred to as ‘swelling-shrinking’ behavior [91,94].

Factors that can affect the LCST and UCST values include pendant functional groups, polymer concentration, and polarity of the medium and molecular weight of the polymer [95]. Since the temperature range from normal physiological sites of the body (37 °C) to diseased sites (40–42 °C) is narrow, thermo-responsive carriers should be able to undergo phase transition precisely. This is important to avoid advanced release of drugs at normal body temperature [96].

Out of various temperature sensitive polymers, poly (N-isopropyl acrylamide) or PNIPam is the most studied thermo-negative polymer. PNIPam is a non-ionic polymer that is synthesized by radical polymerization of N-isopropyl acrylamide. The LCST value of PNIPam is around 32 °C, closer to the normal body temperature. An adjustment in its phase transition temperature can be achieved by copolymerizing with other polymers. Hydrophilic monomers like acrylic acid cause the temperature to increase while a hydrophobic monomer decreases the temperature [94]. Fu et al. synthesized a semi-interpenetrating network via a free radical polymerization process. Upon increasing the acrylic acid concentration beyond 5.5%, LCST of PNIPam increased to 41 °C [97].

PNIPam has the disadvantage of not being biodegradable. Polymers like polyethylene glycol (PEG) could be a useful alternative due to better biocompatibility [98]. For example, Hu et al. carried out research work for evaluating the potential of PLA/PEG based micelles as thermo-sensitive targeted delivery of the anti-cancer drug curcumin. ATRP was implemented for the synthesis of amphiphilic triblock copolymers. The drug was entrapped using the membrane hydration method. Drug release was studied below and above LCST and the release profile was compared with previously reported results of PNIPam based micelles. According to the results, PEG based micelles showed a broader phase transition than PNIPam based micelles. The drug release profile in both cases was faster above LCST. However, the drug release rate was slower in PEG based micelles which is a desired characteristic for controlled delivery in treating cancer [99].

Natural polymers, e.g., hyaluronic acid (HA), chitosan, alginate and dextran, etc. can also be used owing to non-toxicity, good biodegradability, and biocompatibility [98]. For example, *κ*-carrageenan polysaccharide-based thermo-responsive nanogels were synthesized by Danield-Silva et al. using methylene blue (MB) as a model drug. Their results showed that an increase in temperature from (25 °C to 37 °C) and 45 °C resulted in swelling of the nanogel followed by the release of MB [100].

Thermo-responsive nanocarriers have extensive applications in the field of tumor chemotherapy. Thermodox, a thermo-responsive nanocarrier is already in clinical trials for the treatment of breast cancer [96]. Core shell thermo-responsive drug delivery systems can be utilized for overcoming the insolubility issues of hydrophobic and anti-cancer drugs. These nanocarriers have a temperature sensitive shell with a hydrophobic core like polystyrene that acts as a reservoir for loaded drug [98].

Wang and co-workers synthesized a PNIPam based thermo-responsive nanocarrier system for mitochondria-targeted delivery using Paclitaxel (PTX) as a model drug. They also used a non-thermo-responsive PAM (propylacrylamide) based system as control. Since the temperature of mitochondria is high in cancer cells, their results showed an enhanced release profile of drug from PNIPam-PTX system evidenced by better colocalization of PTX in mitochondria of MB49 cancer cell line, whereas PAM-PTX failed to release drug in mitochondria with poor colocalization of the drug. They also stated that the developed nanoparticles were more cytotoxic to the cancer cells in comparison to free drug and PAM based non-thermo-responsive control [101].

In another investigation carried out by Ghamkhari et al., novel thermo-responsive star like micelles were developed using hyperbranched aliphatic polyesters poly(ε-caprolactone)-b-poly(N-isopropylacrylamide) (HAPs-g-PCL-b-PNIPAM) via ring opening polymerization and RAFT techniques. They used docetaxel (DTX) as a model drug to overcome the loading and pharmacokinetics issue associated with the drug. Release profile of the developed system showed an increase in release with an increase in temperature. MTT assay, intracellular uptake and DAPI staining confirmed that the prepared micelles with loaded DTX had significant pharmacokinetics and cytotoxicity in breast cancer cell line (MCF7) compared to free DTX [102].

### 3.5. Light-Responsive Nanocarriers Applied for Diagnosis and Treatment of Cancer

Light as an external stimulus has grabbed considerable attention because of high spatiotemporal precision. Light responsive polymers are non-invasive and release cargo on-demand. Upon exposure to high radiation (ultraviolet, near-infrared, visible) from an external source, these nanocarriers release the encapsulated agents. Generally, these light responsive carriers can be prepared by introducing a photo-cleavable linker or a chromophore as a light responsive moiety into the polymer backbone or matrix of the nanocarriers. Under the irradiation of optimum wavelength, intensity, and exposure time, these photo-cleavable molecules undergo photochemical reactions. These light induced reactions do not require the prerequisite of chemical changes in the environment and can be categorized into (a) photo-isomerization; (b) photo-cleavage; (c) photo-dimerization (d); photo-rearrangement; or (e) photo crosslinking [21,103,104,105,106,107].

Various chromophores have been studied but certain chromophores, e.g., azobenzene [108], spiropyran [109], spiroxazine [110], and nitrobenzyl [111] are considered more efficient than others. In azobenzene, changes in the molecular symmetry occur when the thermally stable trans orientation converts to a less stable cis form. In spiropyrans, irradiation induces a ring opening reaction. UV absorption results in the reversible isomerization of cis to trans form of the photo sensitive groups in nanocarriers which is converted back to cis form by the visible light. Hence, results in disruption of the carriers occur resulting in drug release [112]. Various nanocarriers, e.g., micelles, liposomes, polymeric nanoparticles, hollow metal nanoparticles, etc. are being utilized in photochemical reactions for targeted release of therapeutic agents [113,114]. Additionally, the process of photo-isomerization which is reversible and reproducible functionalizes the nanocarriers as an ‘on-off switch’ [115]. The safety profile and efficacy of a light responsive nanocarrier are affected by the wavelength and power of the irradiation. Hence, the photo-toxicity and penetration depth of light should be taken into account. Generally, the light with a high wavelength results in deeper penetration through skin. For example, according to research, a light at 360, 700 and 1200 nm penetrate 190, 400, and 800 μm, respectively into the skin [116].

Based on wavelength, non-ionizing light can be categorized into three:(a)Ultraviolet light (UV)—200 nm to 400 nm(b)Visible light (Vis)—400 nm to 700 nm(c)Near-infrared light (NIR)—700 nm to 1000 nm

Among these regions, light responsive drug delivery systems mostly respond to UV light because of two main reasons: (i) sensitivity of light responsive materials towards UV; (ii) ability of UV to provide sufficient energy for triggering photochemical reactions. However, UV light suffers from poor penetration and high toxicity rendering the drug release inefficient along with tissue damage [117,118,119]. Light energy depends upon per-photon energy which is inversely related to the wavelength of light. UV light has high energy per photon along with high tissue absorbance, hence a low MPE (maximum permissible exposure) that makes it unsuitable for most clinical applications [118]. On the other hand, NIR and partially visible light have low energy per photon. Their high MPE with high tissue penetration depth due to decreased attenuation with minimum damage to healthy cells making them more suitable for clinical applications [116,120]. NIR responsive nanocarriers are based on three mechanisms; Photo-thermal effect is the most widely studied drug delivery system due to tunable and flexible properties. Metal sulfides/oxides, gold, and carbon nanomaterials are common photo-thermal agents. Two-photon absorption drug delivery systems impart higher excitation while overcoming low penetration issues associated with UV responsive DDS. Up-conversion nanoparticles (UCNP) are nano-scale particles that are inorganic and crystalline in nature that converts NIR excitation to UV emission i-e photon up-conversion. The decreased light scattering results in deeper penetration of biological samples [121].

One drawback with NIR light is only a few compounds respond to this light as NIR is unable to provide sufficient energy for triggering photo-responsive reactions. To overcome this issue, nanomaterials are being formulated that are capable of converting low energy NIR to high-energy UV photons. This results in efficient drug release encompassing two-photon absorption process and up-converting using up-conversion nanoparticles [120]. Light responsive nanocarriers have high potential as drug delivery systems. These carriers could be utilized for tumor therapy guided by imaging as well as in theranostics. Exploiting the photo-thermal effect and generation of reactive oxygen species triggered by light can be a useful ablation of cancers. When combined with other anti-cancer therapeutics, they can be implemented in multimodal cancer theranostics. They have also proven to be highly effective in MDR cancers [122].

Tong et al. developed a photosensitive nanoparticle based drug delivery system using spiropyran as chromophore and UV light a source of irradiation. This triggered on-demand drug release as well as enhanced tissue penetration because of reversible change in the volume of particles [123]. Yan and coworkers addressed the drawback of light responsive drug delivery systems that require UV/Vis excitation, by demonstrating an efficient strategy. Making the use of continuous wave diode NIR laser showed NaYF4: TmYb UCNPs encapsulated in block copolymer micelles emitted photons in UV region upon exposure to 980 nm light. O-nitrobenzyl groups resulting in activation of photo-cleavage reaction absorbed these photons. This led to the disruption of block copolymer micelles and thus release co-loaded agents [124].

In another investigation, Luo et al. reported the development of long circulating nanoparticles that demonstrated the ability of releasing drug upon irradiation. They established a systematic approach for designing stealth liposomes with porphyrin-phospholipid (pop) using doxorubicin as the therapeutic agent. NIR was used for triggering the release of a drug. The developed delivery system exhibited enhanced stability and extended circulation time in blood. They stated that chemo-phototherapy with pop stealth liposomes showed far more efficacy than conventional phototherapy [125]. In a study conducted by Croissant et al., mesoporous silica nanoparticles based two-photon triggered drug delivery system was developed using azobenzene and two-photon fluorophore. At the low power of laser, the florescence of fluorophore resulted in efficient two-photon imaging of the cancer cells. At high power and a short duration of exposure, the nanovalves exhibited two-photon triggered release in cancer cells [126].

### 3.6. Advancement in Endogenous Stimuli Sensitive DNA Based Smart Nanocarriers

Several endogenous stimuli in pathological environments including temperature, low pH, oligonucleotides can be applied for particular triggers. As a result of malignancy, cell proliferation results in imbalances in nutrient, oxygen levels. The relative differences in pH between the extracellular and intracellular cancer cells are the most distinguish pathophysiological feature [127].

#### 3.6.1. pH Responsive Cancer Targeting

Various DNA-assisted and pH-responsive drug delivery systems have been identified by the researchers. Several researchers studied the pH sensitive i-motif structure DNAyzme and structure stabilization. The i-motif is a motif that can be used in a variety of in acidic environments; DNA has structures that form stable links between anti-parallel, cytosine-rich four-strand sequences, forming the tetraplex structure through C base protonation, which favors interactions with other cytosine bases over guanine. The nucleic acids are made up of a duplex structure called nucleic acid bridges, which is made up of the i-motif and its sequences [128]. Rolling circle amplification was created by Tian et al. to produce polymeric DNA composed of tandem units of functional sequences. The i-motif forms a structure and releases the drug to cause apoptosis when exposed to acidic conditions. Wang et al. developed a pH-responsive anti-cancer drug delivery system using a self-catalyzing DNAyzme and a rolling circle amplification method [129].

Coated polymer/DNA nanocomplexes containing a high mobility group box 1 (HMGB1) were developed as a competent non-viral gene delivery system by Mingyue Wang et al. Nanocomplexes with a pH-sensitive core shell system have been formed and characterized. Free folic acid blocked gene transfection and expression in KB cells, according to the findings. The developed nanocomplexes showed enhance fluorescence protein expression at the tumor site [130].

Olcay Boyacioglu et al. created a DNA aptamer to prostate specific antigen with fixed sequences to facilitate Dox binding and dimeric aptamer complexes. The cellular was directly internalized by prostate-specific membrane antigen (PSMA+) cancer cells. Dimeric aptamer complexes (DACs) are complexes that carry Dox to PSMA+ cancer cells. Under physiological conditions, Dox was released from the DAC-D with an 8-h half-life. Dox was delivered to C4-2 cells using DAC-D with nuclear localization and endosomal release. DAC-D has specificity and durability, which could help with Dox delivery to tumor tissues in vivo [131].

Nanoparticles made of polyethyleneimine (PEI) and a pH-sensitive diblock copolymer were formed by Sethuraman et al. Due to the shielding of PEI by poly(methacryloyl sulfadimethoxine) PSD-b-PEG, the nanoparticles containing DNA/PEI/PSD-b-PEG were small and had low cytotoxicity at pH 7.4. PSD-b-PEG attached to the PEI/DNA complex reduced the interaction of PEI positive charges with cells by 60% and reduced cytotoxicity. At pH 6.6, the nanoparticles showed increased cytotoxicity, indicating PSD-b-PEG detachment from nanoparticles, allowing PEI to attach to cells. The following forms of nanoparticles can distinguish minor pH differences between normal and tumor tissues and have a lot of potential for targeting tumor tissues [132].

#### 3.6.2. Oligonucleotide Responsive Nanocarriers

There are a large number of applications of oligonucleotides (microRNA and small interfering RNA) in tumors. Oligonucleotides such as siRNA and microRNA are active agents that have been used for active drug delivery at the malignant site. Nanoparticles are applied to deliver oligonucleotides at malignant sites. The application of iron oxide, gold and quantum dots ligated with contrast agents has facilitated the early diagnosis and analysis of therapeutic efficacy. By strand displacement, the nano-carriers can be reconfigured and released. A single stranded oligonucleotide that is complementary to the region of double stranded DNA is used to rehybridize and dehybridize the double stranded DNA [133]. One group of researchers created an oligonucleotide-responsive DNA nanosuitcase that encapsulates siRNA by connecting two opposite DNA and siRNA end terminals in a complementary manner. Under biological conditions, the targeting moiety within the nanocarrier was covered, but it was released when an oligonucleotide trigger, such as miRNA or mRNA, was recognized. Li et al. created a nanocarrier with DNA and multilocked DNA valves for mRNA-responsive drug delivery. The researchers encapsulated Dox in mesoporous nanoparticles, which were then capped with two gate DNAs via electrostatic interactions. These DNAs were found to be complementary to tumor-associated GT mRNAs or Tk1 [134]. The cargo can be released by nanoparticles in cells that overexpress mRNAs. Shi et al. created DNA nanoflowers with MUC1 apartmers for tumor targeting and anti-miR-21 for miR-21 responsive release. For CRISPR/Cas9 genome editing, the DNA nanoflower was encapsulated with Cas9/sgRNA into nanoflower through hybridization between the stem loop of the sgRNA and the anti-miR-21. When tumor cells were incubated with a miR-21 mimic resulting from miR-21 responsive Cas9/sgRNA release, the genome modulating efficiency was increased [134,135].

#### 3.6.3. Multiples and Molecular Biomarker Responsive Nanocarriers

For more specific targeted drug delivery, the researchers have developed a delivery system with more than two stimuli. In which activation of the responsive moiety is compulsory for the release of the loaded compound. A group of researchers developed a mesoporous silica nanoparticles, that are dual responsive to enzymes and biomarkers for controlled release of drugs and also for dye [135]. Another group of researchers developed a DNA conjugated gold nanoparticles that disassembled in result to low pH and specific enzyme for tumor associated drug delivery. In this study, the pH and telomerase stimulated thiolated DNA were absorbed onto gold nanoparticles via Au-S binding that results in the assembly of nanocarriers at physiological pH. Moreover, it can cause the disassembly in the tumor environment via pH responsive triplex structure formation. Zhou et al. developed mesoporous silica nanoparticles that are triggered via redox reactions, enzymes and heat [136]. The calcein was encapsulated in the capped and pores via self-complementary duplex DNA [137].

The loaded compound was released after denaturation by DNase and bond cleavage by disulfide reducing agents such as dithiothreitol or glutathione. Biomolecules such as ions, protein, and small molecules are recognized as potential triggers for controlled release in drug delivery applications because of their increased bioavailability at the disease site. ATP was utilized as a trigger mechanism for drug release through conformation reconfiguration [138]. These locks are made up of various aptamer combinations that recognize single or double biomarkers expressed in tumor cells. When the biomarker was bound to both locks, the locks were unfastened, and the origami box unlatched and released the filled compound thermodynamically. Liu et al. designed and introduced a doll-like DNA nanocage with DNA tetrahedra of different sizes but similar structures for ATP sensitive disassembly. Each layer was hybridized with an anti-ATP aptamer and its complementary sequence, and the small tetrahedra were sequenced with larger tetrahedra. Figure 4A schematic representation of different types of exogenous and endogenous stimuli Figure 4B showed various types of stimuli responsive nanoparticle for tumor targeting. Aptamer adhesion was engineered to be preferable to duplex formation. As a result, in the presence of ATP, the hybridized tetrahedra dissociate, resulting in the isolation of the tetrahedral structure [23,139,140].

#### 3.6.4. Redox and Enzyme Responsive Smart Carrier System

Enzymes play key functions in a number of disease states and many of them catalyze the breaking of the particular peptide bonds. The substrates of these enzymes are present at the surface in the cytoplasm or within various cellular organelles. These tumor-associated enzymes are connected to different key events including tumor progression, tumor growth, extravasation and metastasis. The enhanced levels of particular enzymes including glycosidases, proteases, and phospholipases are signals of various types of tumor cells. Many enzyme responsive delivery systems explore the outside the cell environment [141,142]. Metalloproteinases (MMPs) are the most trigger for controlled drug release. These MMPs are over expressed in the extracellular environments in many kinds of tumors. Singh et al. synthesize a stimulus responsive system based on polymer coated mesoporous silica nanoparticles that encapsulate drug into both shell and core domains. Another researcher group developed a class of multifunctional type nanoparticles to achieve stimuli responsive targeting drug delivery. However, anti-cancer drugs could be effectively encapsulated in the nanoparticles and produced the cell death of MMP tumor cells. There are some intracellular enzymes including cathepsin B, elastase or glycosidases are also exploited for controlled drug release. Cathepsin B is a lysosomal protease that is responsible for cancer cell progression with a particular peptide. Therefore, it gives an attractive option for triggering specific cancer targeting. The differences in reduction efficiency between tumor and normal tissues between extracellular and intracellular environments can be useful for targeted release at the malignant site [143,144]. The GSH concentration is very low in the extracellular environment but is concentrated within the cell inside the cytosol. These differences are more visible in tumor tissues. Wu et al. synthesis a biocompatible and biodegradable 1-cysteine based poly(disulfide amide) for fabricating reduction sensitive nano-carriers with greater hydrophobic drug encapsulated properties. The following GSH sensitive crosslinking agents can also be encapsulated either inside the shell or in the core of micelle-based nanoparticles [145,146].

#### 3.6.5. DNA Based Hybrid Nanocarriers System for Cancer Targeting

Nanoparticles have greater potential to achieve dual functionality if more than one type of nanostructure can be encapsulated in a nano-assembly referred to as hybrid nanoparticles. The recent developments in the synthesis and evaluation of hybrid nanoparticles are based on two design techniques in which micellar, porous silica, viral, noble metal and nanotube systems are incorporated within or on the surface of a nanoparticle. Self-assembled particles comprised of the oligonucleotide, block copolymers of poly-(ethylene glycol)-block-poly(aspartic acid) were developed by the mixing of calcium/DNA and phosphate/PEG-PAA solutions. The particles have the ability to encapsulate DNA in the core with good efficiency as determined by fluorescence and gel permeation chromatography. Moreover, the cytotoxicity of particles evaluated by MTT assay. The following organic-inorganic hybrid nanoparticles encapsulating DNA components that utilized as DNA delivery systems for gene and antisense therapy. Judy M. Obliosca et al. developed Au-DNA conjugates via equilibrating phosphine stabilized gold nanoparticles. The modified glass substrates method was used to immobilize onto MPTMS (3-mercaptopropyltrimethoxysilane). The AuNPs quenching efficiency with increased Au-to-dye was assessed using ligand exchange of thiolated oligo nucleotide via 2-mercaptoethanol. The UV-vis fluorescence and absorption based effective exchange was achieved in few minutes. The findings showed that by using array format the fluorescence quenching of cy3 Au-DNA can be assessed. Maryna Perepelyuk et al. develop the mucin1-aptamer functionalized miRNA-29b-loaded hybrid nanoparticles in lung cancer mice. The efficacy of developed hybrid nanoparticles to down regulate oncoprotein at the cellular and in vivo was observed using Western blot and tumor growth was observed using bioluminescence. Results show that the Muc1-aptamer conjugated to the surface of the nanoparticles increased the delivery of miRNA-29b to tumor cells. Moreover, the down-regulation of DNMT3B by MAFMILHNs showed the inhibition in transgenic mouse model. Yoshinori Kakizawa and Kazunori Kataoka et al. developed self-assembled nanoparticles developed by calcium phosphate, block copolymers, oligonucleotide via simple mixing of phosphate/PEG-PAA and calcium/DNA solutions. The copolymerization of PAA and PEG segments is necessary to avoid the precipitation of calcium phosphate crystals to form nanoparticles. The nanoparticles can carry the ability to encapsulate the DNA in the core with enough efficiency as analyzed via gel permeation and fluorescence measurements. The following organic-inorganic hybrid system encapsulating DNA molecules can be used as delivery system for gene therapy [147]. Corey J. Bishop et al. developed hybrid gold nanoparticles (AuNPs) coated with layer-by-layer polymer coatings to enable the co-delivery of DNA and siRNA. When hybrid polymer/nucleic acid/gold nanoparticles were added to the brain cancer cells, the hybrid nanoparticles are taken up by the cells and evaluated by exogenous gene expression. The polymer/nucleic acid and AuNPs hybrid nanoparticles are theranostics platform that can deliver the genetic therapies to human cells [148].

Surface functionalized hybrid mesoporous silica nanoparticles are efficient and safe carriers for bioactive molecules. The non-covalent binding of polyethyleneimine (PEI) (C_2_H_5_N)_n_ polymers to the surface not only enhanced the cellular uptake but also produce a cationic surface to which DNA and siRNA construct could be ligated. The particles coated with PEI polymer were efficient for transducing HEPA-1 cells with siRNA that showed the ability to knock down GFP expression. The results demonstrated the enhanced cellular uptake of the silica nanoparticles to the pancreatic tumor cells. It was demonstrated that with the careful selection of PEI size it might be possible to develop cationic mesoporous silica nanoparticles that have the ability for enhanced drug delivery with low cytotoxicity [149].

Fujian Huang et al. developed aptamer coated PLGA hybrid nanoparticles core shell lipid polymeric structures through nano-precipitation and self-assembly. The aptamer sgc8, can bind to human protein tyrosine kinase 7 overexpressed on target CEM (human leukemia cancer cell line) cell membranes. The sgc8 aptamer was then designed to hybridize with a diacyllipid-modified DNA strand via tail with repetitive 5′-GCA-30 sequences. After synthesizing hybrid particles, we validated the selective binding of the NPs to the targeted cells. For effective drug delivery, NPs must be internalized by pathogenic cells. In comparison to the previous studies, the data revealed that a greater TAMRA fluorescence signal in the cells showed that TAMRA labeled nanoparticles were specifically internalized into targeted cells. The cellular uptake studies showed that the NPs can be taken up by targeting cells. The cytotoxicity results showed that this targeted delivery system enhances the anti-tumor efficiency. Different nanoparticles can be applied for the targeted delivery of other hydrophobic drugs. Figure 5 shows hybrid nanoparticles types for tumor targeting and diagnosis [150].

## 4. Various Synthesis Strategies for Smart Nanocarriers System Applied for Diagnosis and Treatment of Cancers

DNA is the basic building block of life, through its unique nature to pair up with complementary base sequences (adenine-thymine and guanine-cytosine), it can build up multifarious DNA-based nanostructures. Properties like self-assemblance and recognizing specific molecular sequences enable DNA-based nanocarriers to be used in various biomedical applications, disease therapy, drug delivery, imaging, diagnostics, and theranostics. Moreover, induction of stimuli-sensitive characteristics endures DNA nanostructures highly capable to respond and work under a specific stimulus only. Therefore, such types of smart DNA nanostructures are specifically fruitful in site-specific cancer therapy and diagnosis. Synthesis of stimuli-responsive DNA nanocarriers based on physical, chemical, and biological linking of different moieties to various DNA template strands [23]. The principle involved either self-assembly, hybridization to complementary base or attachment through linkers, or physical adsorption or intercalation. Some of the synthesis strategies for stimuli-sensitive DNA nanocarriers are discussed below.

### 4.1. Synthesis Strategy of Self-Assembled DNA Hydrogel

Enzyme-responsive self-assembled DNA hydrogel was developed by Li et al. to control hydrogel size and mediate stimuli-responsive targeted drug release. Mainly, the synthesis involved two Y-shaped DNA building blocks with sticky ends namely Y-shaped monomer A with three sticky ends and Y-shaped monomer B with 1 sticky corner [151], while a DNA linker with two sticky ends was used to link the two monomers. Moreover, the other end of Y-shaped monomer B was additionally tuned with different functional units, therefore, monomer B possessed the dual role of blocking unit, prevented the extension of a nanostructure, and of targeting unit because of a functional moiety that recognizes cancer cells. Functional moieties like aptamers, antisense oligonucleotides, and DNAzymes that are used for specific targeting of cancer cells, inhibition of cancer proliferation, and prevention of cell migration, respectively, were attached to building block monomers, specifically B, to induce specific qualities to the monomer.

Both Y-shaped DNA monomers, A and B, through their sticky ends hybridized with the complementary sequence on the linker to form DNA nanohydrogel. Further, to incorporate stimuli-responsive properties, the two monomers and the linker were introduced with disulfide linkage that resulted in stimuli-responsive DNA nanohydrogel. Di-sulfide linkages made DNA nanohydrogel stable in the blood circulation, while the glutathione (GSH) reducing enzyme, present in the cytosol of cancer cells, easily cleaved it. Thus, GSH enzyme-sensitive DNA nanohydrogel released a payload of the therapeutic (aptamers/antisense oligonucleotide) genes that regulate various functions specifically inside the tumor cells. Moreover, by adjusting the ratio of two monomers, the size of the nano-hydrogel can be controlled according to the desired function. For instance, DNA-aptamer nanohydrogel successfully prevented proliferation and migration of specific human lung adenocarcinoma epithelial (A549) cell lines, not affecting the control cells [151]. The system was flexible in design, biocompatible, able to incorporate various therapeutic genes, thus, promising for targeted gene delivery for the treatment of a variety of cancers.

### 4.2. Synthesis Strategy of DNA Origami-Based Nanostructures

DNA origami is a self-assembled nanostructure having nanometers to the sub-micrometer size range. It involves DNA folding to two- or three-dimensional nanostructures. The fabrication of DNA origami was based on the folding of DNA scaffold, which is a long single-stranded DNA along with many short ssDNA, the staples [152]. DNA origami-based synthesis facilitated easy production, robustness, increase product yield, and provide an opportunity to form complex shapes [153]. DNA origami provides a platform for automated fabrication of nanostructures, bioimaging, biosensors, and drug delivery for cancer therapeutics [154].

For instance, DNA origami-based synthesis enables control over the size, shape, and dimensions of DNA nanostructures and enriched with various functional groups to facilitate biomedical applications and drug delivery. A DNA nanorobotic system, based on DNA origami, was synthesized to deliver thrombin to the cancer cells [155]. Thrombin delivery to tumors caused blood vessel thrombosis, thus, suppressing the supply of oxygen and nutrients that lead to tumor cell death. The synthetic scheme involved the assembly of single-stranded M13 bacteriophage DNA with multiple staple strands to produce a rectangular-shaped (90 nm × 60 nm × 2 nm) DNA origami sheet. To load thrombin, staple poly-A oligonucleotides strands were extended at four distinct locations on DNA origami sheets. Furthermore, thrombin was modified to be anchored into DNA origami sheets through linking thiolated poly-T oligonucleotides to the thrombin molecules by using the cross-linker, sulfosuccinyl-4-(N-maleimidomethyl) cyclohexane-1-carboxylate. Then, modified thrombin was anchored through hybridization with poly-A extended strands of DNA origami sheets. To achieve site-specific delivery, DNA nanorobot was multi-functionalized with DNA aptamers (AS1411). Thus, conjugation of aptamers to DNA origami sheets formed tube-shaped DNA nanorobot. In the tumor cells, aptamers specifically targeted nucleolin, the tumor-associated endothelial cell protein. The aptamer-nucleolin interaction produced a mechanical movement that served as a stimulus to open DNA nanorobot and to release thrombin at the tumor site [155].

### 4.3. Synthesis of Mesoporous Silica (M-SiO_2_)-DNA Nanocomplexes

Recently, the focus has been laid on the fabrication and utility of combined M-SiO_2_-DNA nanocomplexes. Because the nanocomplex has multiple benefits such as the production of bulky DNA nanostructures, capping of M-SiO_2_ pores by efficient DNA nanostructures, application of different stimuli like temperature, enzymes, pH, ligands to release the drug cargo by opening the pores due to reconfiguration of the capping units, and tailoring of M-SiO_2_-DNA nanocomplexes with cancer-specific ligands like aptamers that mediates targeting of cancer cells only [156].

In an attempt, stimuli-responsive DNA capped M-SiO_2_/Fe_3_O_4_/AuNPs were fabricated for the delivery of DOX to the cancer cells. The fabrication scheme involves at first preparation of Au and magnetic Fe_3_O_4_ NPs, then deposition of AuNPs on octahedral magnetic Fe_3_O_4_ NPs to form core–shell trisoctahedral Fe_3_O_4_/AuNPs. Afterward, it was further coated with the M-SiO_2_ layer and chemically modified with aminopropyltriethoxysilane layer. The next step was to functionalize NPs with maleimide. A double-stranded oligonucleotide was prepared side by side. DOX was engraved in M-SiO_2_ pores through covalent bonding of double-stranded oligonucleotide to the maleimide group. The linkage is thermo-responsive; therefore, exposure to near-infrared light (NIR) unblocked the pores and released the drug in a controlled manner at the cancer cells. Additionally, DNA-capped M-SiO_2_/Fe_3_O_4_/AuNPs were driven by the magnetic field because of magnetic metallic elements in the nanosystem. Overall, the combo of photothermal (NIR) and magnetic stimuli guided the DNA capped M-SiO_2_/Fe_3_O_4_/AuNPs to target and deliver the drug only to the cancer cells, subsiding the chance of mistargeting the normal cells [157].

### 4.4. Synthesis Strategy of Smart Functionalized DNA Supramolecular Nanostructures

Another approach is to synthesize stimuli-response DNA-based supramolecular nanostructures. A supramolecular nanostructure was based on non-covalent interactions of the molecules, resulting in a large variety of spherical, rod-shaped, crystal-shaped, origami-like, sheet-like, micelles-like structures. i-motif quadruplex DNA is a supramolecular nanostructure that was used to cap metallic nanoparticles to form stimuli-sensitive and controlled release quadruplex DNA-metal nanocontainers. i-motif is basically a four-stranded DNA with cytosine base stretches and undergoes structural conformation change on a pH change. Therefore, the i-motif DNA cap acted as a gate to open and close the pores of M-SiO_2_ in response to specific pH stimulus that can be exploited for cancer drug delivery. The gate on/off was controlled and reversible to allow drug release up to the pre-determined extent [158]. At a lower pH, the cytosine was protonated and presented a C-tetrad form of i-motif structure, while at a high pH, cytosine was deprotonated to unfold into single-stranded DNA. For instance, dual chemo-photothermal-responsive aptamer functionalized i-motif DNA-gold (aptamer-i-motif DNA-Au) nanoconjugates were developed to achieve a high targeting ability to treat cancer cells [159]. Synthesis steps were: (1). Preparation of AuNPs, (2). dsDNA was coated on AuNPs through thiol groups on the Au surface in the presence of NaCl, (3). Then, targeting ligand, aptamer (AS1411) was integrated with i-motif strand (s2), (4). DOX was intercalated into C-G base pairs of double stranded DNA through hydrophilic and hydrophobic interactions. Under normal pH, single-stranded s2 hybridized with s1 complementary sequence to form double stranded DNA, while after endocytosis (lower pH), s2 form stable conformation and de-hybridized to trigger drug release. On the tumor site, aptamer-nucleolin interaction mechanically stimulated the nanoparticles and both tumor microenvironment lower pH and photothermal-NIR irradiation were the stimuli behind drug release from this smart nanosystem [159]. Similarly, researchers developed AuNPs with DNA-capped surfaces. AuNPs were modified via two types of oligodeoxynucleotides, i.e., i-motif binding oligodeoxynucleotides and BCL-2 antisense oligodeoxynucleotides. At the acidic pH of cancer cells, cluster disassembly leads to DOX drug release [160].

### 4.5. Synthesis of Deoxyriboszymes (DNAzymes) Based Nanostructures

DNAzymes are ssDNA molecule with well-defined catalytic properties, economical to produce, selective, and direct site-mediated gene silencing as cancer therapeutic. The main functions involved mRNA cleavage, ligation, or DNA phosphorylation [161]. DNAzymes were modified to respond to various stimuli like pH, temperature, photothermal, small molecules, enzymes, etc. [161].

Resistance against chemotherapy and radiotherapy limits triple-negative breast cancer therapy. To overcome the issue, DNAzyme modified M-SiO_2_ coated Au nanorods that were responsive to photothermal NIR stimulus were fabricated [162]. The incorporated survivin DNAzyme was linked to M-SiO_2_- Au nanorods through the thermal-sensitive small molecule, 4,4′-azobis(4-cyanovaleric acid) (AC). Survivin is mainly involved in tumorigenesis and the proliferation of cancer cells. To fabricate, at first, M-SiO_2_- Au nanorods were prepared with NH2 functionality. Then, carbodiimide crosslinker, EDC/NHS chemistry, was used to modify the surface of AC. Finally, DNAzyme was chemically bonded to the prepared nanorods via AC. The designed regime worked to improve breast cancer cell sensitization. Upon irradiation, NIR absorbed by the gold nanorods converted into heat energy which broke the bonds and released DNAzyme to downregulate survivin mRNA and improve cancer cells (MDA-MB-231) sensitization towards photothermal therapy [162].

### 4.6. Synthesis of DNA Nanowires

Recently, stimuli-responsive DNA nanowires were constructed to provide combined photodynamic and chemotherapy and to specifically targeted cancer cells. For synthesis, the building blocks comprising of two short DNA chains were self-assembled via super-sandwich hybridization reaction to form DNA nanowires of 500 bases and 167 nm length. Further, photosensitizer, chlorine e6 (ce6) were linked to nanowires through covalent bonds, and doxorubicin was loaded through non-covalent intercalation interactions. Efficient photodynamic therapy by the DNA nanowires in conjunction with chemotherapy produced reactive oxygen species and kill HepG-2 cancer cells [96].

### 4.7. Synthesis of Smart DNA-Lipids Nanostructures

Stimuli-sensitive DNA-lipids nanostructures were produced cancer-cell-specific targeting. For instance, aptamer was derived from a 26-mer DNA aptamer (AS1411) to be used as a DNA-based ligand covering the liposomes and formed N-APT-liposomes [163,164,165,166,167,168,169,170]. This derived aptamer (N-APT) had a high affinity for nucleolin, which is involved in cancer cell proliferation. Thus, nucleolin served as a tumor microenvironment endogenous stimulus to interact with aptamer and then configured the N-APT-liposome structure to release the payload. The synthesis started with tagging of N-APT with cholesterol at the 3′ end. Then, cholesterol tagging inserted the aptamer into the hydrophobic lipid membrane and immobilized it on the liposome surface. Then, liposomes were prepared through the thin film hydration method, using components like cholesterol, hydrogenated soy phosphatidylcholine, and disteroyl phosphatidylethanolamine with attached methoxy poly (ethylene glycol) moiety (mPEG2000-DSPE). To conjugate N-APT to liposomes, cholesterol tagged N-APT was added to the above reagents mixture in the hydration phase of the thin-film hydration method. The prepared N-APT-liposomes were loaded with a fluorescent dye or drug and tested on breast cancer (MCF-7) and prostate adenocarcinoma (LNCaP) cell lines [163]. All DNA-based nanostructures synthesis strategies are summarized in Table 3.

## 5. Mechanism or Cellular Internalization of Smart Nanocarrier System Applied for Diagnosis and Treatment of Cancers

DNA molecules are negatively charged because of the presence of phosphate groups, therefore, faced electrostatic repulsion from the anionic cell membrane. Therefore, it is difficult for DNA-based nanostructures to internalize directly through the cell membrane. However, DNA nanostructures have internalized intracellularly because of the energy-dependent endocytosis process. Depending on their sizes, different types of endocytotic pathways carried DNA nanostructures inside the cells including phagocytosis (>0.5 µm size), caveolin-mediated endocytosis (60 nm), clathrin-mediated endocytosis (120 nm), macropinocytosis (>1 µm), and caveolin- and clathrin-independent pathways (90 nm) [171]. Moreover, the precise shape and geometry of DNA-based structures facilitated internalization. Since the curvature of lipidic membranes plays a major role in vesicles fusion and budding in the endocytosis and proteins governed their shape and orientation. DNA nanostructures resembling membrane-sculpturing proteins (SNARE, clathrin, dynamin, etc.) could direct the cell membrane curvature and bending that could manipulate the endocytosis process [172,173,174]. It has been reported that cholesterol-modified DNA origami nanostructures could be unfolded in the presence of lipid bilayer membrane or surfactant and hydrophobic interactions governed their modification [175]. Further, modification of DNA nanostructures with cationic polymers like PEI can facilitate cellular uptake [176]. Moreover, the shape and base sequence of the DNA nanostructures also determined the endocytosis mechanism. For instance, spherical nucleic acid nanoparticles were endocytosed through a lipid-raft-dependent, caveolae-mediated pathway [177]. DNA tetrahedrons also exhibited caveolin-mediated endocytosis and were then transported to the lysosomes through a microtubule-dependent way [178]. Next, it was observed that DNA origami nanostructures’ cellular uptake efficiencies were largely determined by their shape, large size and high aspect ratio lead to enhanced uptake. Scavenger receptors were pivotal in their uptake; however, the mechanism is not clear in H1299 and DMS53 cells, while it was predominantly caveolin-mediated endocytosis in HeLa cells. Additionally, these DNA origami nanostructures are finally disposed to endosomes and lysosomes in microtubule-dependent manner [179]. Functionalized/ligand anchored DNA nanostructures adopted receptor-mediated uptake. It was observed that the scavenger receptor, oxidized low-density lipoprotein receptor-1 (LOX-1), was responsible for the cellular uptake of biotin functionalized octahedral DNA nanocages [180]. In a study showed that COCO enhances the efficiency of photoreceptor precursor differentiation in early human embryonic stem cell-derived retinal organoids [181]. Aptamer conjugated icosahedral-DNA nanoparticles bearing mucin-1 aptamer have internalization via clathrin-mediated pathway and transported to lysosomes, just like the mechanism of free mucin-1 [182]. Star-shaped aptamer-i-motif DNA-Au nanoconjugates, having AS1411 aptamer, bound to nucleolin that in turn facilitated their uptake. Thus, the nucleolin-mediated endocytosis was the cellular uptake mechanism [159]. Whereas nucleolin-targeting aptamer (AS1411) mediated endocytosis through macropinocytosis, stimulated by nucleolin-dependent mechanism, in a variety of cancer cells and transported to the nucleus [183]. Recently, DNA nanowires for chemo-photodynamic therapy of the cancer were directed largely into the cytoplasm, therefore, the possible internalization mode was the endocytosis-independent process. Probably DNA nanowires bound to the cationic sites in the cell membrane that led to their cellular uptake [96]. The endocytosis-independent provides an efficient and rapid pathway for drug delivery. There is a controversy in determining the final fate of artificial DNA nanostructures from endosome/lysosome to endosomal escape. Moreover, their cellular internalization may involve more than one uptake mechanism [172]. Further, diversity in smart DNA nanostructures in terms of shape, size, gene sequence, configuration, etc. affects the endocytosis pathway. Nevertheless, efforts are being made to improve cytosolic delivery and endosomal escape for the smart DNA-based nanostructures. Figure 6 illustrates the pathways and mechanism of internalization of DNA assessed nanoparticles.

## 6. Potential Immuno-Modulatory Effect of DNA-Based Nanostructures

The immunostimulatory and immunomodulatory nucleic acids are the most common assistant in the immunotherapy of various diseases. The cytosine phosphate guanine (CpG) are able to respond to different Toll-like receptors 9 (TLR9) [184]. The immunomodulatory nucleic acids have been applied to treat cancer. ODN (Oligodeoxynucleotides) containing an unmethylated CpG are considered to be an efficient immunotherapeutic vaccine, in addition, to help achieve efficient therapeutic applications as it can stimulate the Toll-like receptors 9. These mentioned vaccines have been studied and used for melanoma immunotherapy, breast cancer and glioblastoma multiforme [185]. The stimulation of TLR9 can modulate the dendritic cells (DCs), B cells, macrophages to put together pro-inflammatory cytokines. Li et al. reported nanostructure of DNA as the most commonly applied delivery system for DNA tetrahedron, tubular DNA origami and CpG. The released pro-inflammatory cytokines after caught up by cells and identified by TLR9 generate immunotherapeutic effects in various diseases. Huang et al. reported that binding to the TLR9 stimulates an NF-kB signaling cascade to assist the pro-inflammatory cytokines including (TNF-α) tumor necrosis factor-α, (IL-6 and IL-12) interleukin-6 and 12 along with co-stimulation of CD80 and 86 [186]. These cascades of events govern immunostimulatory response while inhibits the Th2 adaptive immune responses. DNA nanostructures are the most promising carriers for immunizing various human diseases including (tuberculosis, hepatitis B, Alzheimer’s disease and malaria etc) [187]. King et al. reported that DNA-based immunization is very successful in stimulating the humoral and cellular immune responses without triggering immunity against the vector. In cancer therapy DNA nanostructures have been designed to stimulate the patient’s immune system to treat tumor cells. DNA originated and isolated from the *Bacillus Calmette-Guerin* and other immunostimulants demonstrated a lot of benefits in the treatment of the malignancies including the bladder. Their immuno modulatroy response activity is dependent on the particular oligodeoxynucleotide sequences comprised of CpG (cytosine-phosphate-guanosine) [188]. CpG sequence can recognize the Toll-like receptors and generate the immune response that served as a stimulant in immunotherapy for cancer targeting. In addition, many proteins such as immunostimulatory monoclonal antibodies (mAbs) can also produce immune responses and stimulation of inflammatory processes that have ability to be applied for the immunotherapy for cancer and several other infectious diseases. The single stranded DNA comprises CpG are not stable in normal physiological conditions. The most commonly applied approaches to enhance the stability of oligo-CpG in physiological conditions is the chemical modification of DNA backbones. The transportation of CpG or CpG modified oligo-CpG into cells via nonspecific endocytosis, limited the immunotherapy [189]. Rattanakiat et al. developed the ligating CpG comprising Y-shaped monomers, dendrimers like DNA [190]. The Fan and co-authors developed self-assembled DNA tetrahedral nanostructures by using four (55-mer strands). The immuno-modulatroy response generated by CpG motif of DNA nanostructures via generating greater level of secretion of various pro-inflammatory cytokines including interleukins (IL-6, 12), tumor necrosis factor (TNF-α) is more significant and stronger than those generated via single-stranded CpG oligonucleotides [191]. The DNA tetrahedral carriers were small in size and are stable under normal physiological conditions and showed no toxicity, representing the promising applications in immunotherapy [192]. Liedl and co-workers developed 30-helix DNA origami nanostructures as the cargo for CpG motif delivery. The CpG decorated DNA origami stimulated stronger immune responses than equal amounts of CpG oligonucleotides linked with lipofectamine, a standard transfecting system. Enhanced cytokine production and immune cell stimulation were achieved via CpG decorated DNA origami [151,193].

## 7. Comparative Studies and Superiority of DNA-Based Nanostructures over Other Carrier Systems in Cancer Targeting

As already mentioned in the previous section that DNA based nanostructures endorsed many unique features and benefits including biocompatibility with zero risk of an immune response, flexibility, and a simple self-assembly process. These features are pre-requisites for effective drug carriers. The wide sticky ends and features spatial structures can be used to deliver therapeutic agents into cells. DNA being a component of the human genome is non-toxic, non-immunogenic, and relatively stable in both in vitro and in vivo studies [194]. The good qualities and immobile linkages among nucleic acid bases make DNA an efficient approach for multifunctional nanomaterials. It is best to be given in cancer as traditional chemotherapeutic agents such as doxorubicin and paclitaxel face great challenges to non-specific distribution along with side effects [195]. The small molecular cytotoxic drugs are usually nonspecifically distributed via blood circulation that can cause severe systemic toxicity. Many broadly used drugs have limited accumulation at tumor sites due to inherent insolubility in aqueous solution. Multidrug resistance in cancer cells is a major factor leading to the failure of many forms of chemotherapy. Therefore, the development of effective and safe chemotherapy drug carriers have been a significant task [196]. Macromolecules and metal based nanoparticles are promising candidates for drug delivery in nano-medicines. Many research investigations and early clinical studies show that the drug loaded nanoparticles exhibit minimum adverse effects due to their optimal cellular uptake. In the case of metal nanoparticles, gold nanoparticles have already been applied for the controllable release of anti-tumor drugs in laboratory research [197,198]. Despite, the biocompatibility and inertness of the metal nanoparticles, safety challenges have to be faced. The metal nanoparticles may stay in the body after drug administration and chronic accumulation can lead to the toxicity. Thus, the optimal drug delivery system should possess the properties for both safety and efficacy considerations. The convenient and easy method of self-assembled method for DNA nanostructures offers a diversified platform for anti-tumor drug delivery [199]. The DNA based drug delivery carrier system have a great ability for control accumulation behavior in the tumor due to shape and size dependent increase permeability and retention effects (EPR). Various surface modifications on the DNA, facilitate tumor targeted delivery with efficient and sustained drug release properties for effective chemotherapeutics. In comparison to other conventional nano-materials, DNA nanostructures have some significant benefits over others carriers systems [200]. The nanoparticle systems have raised concerns regarding possible adverse effects. DNA nanostructures are non-cytotoxic, biodegradable, and biocompatible. The DNA nanostructures and DNA origami assembled into arbitrary shapes and sizes. The DNA surfaces can be modified in a precise manner with molecular accuracy. The folding of the long single stranded DNA scaffold into desired nano-scale shape via hybridization with a set of short staple strands. Each of these staples have a unique sequence and can be addressed and modified to carry different molecules such as proteins, dye and drugs. This approach not only enable the efficient encapsulation of the DNA nanostructures with various therapeutic carriers and also used for targeting binding and facilitate cellular uptake and govern conformational switching in response to external stimuli [196,201].

## 8. Biosafety/Biocompatibility, Stability and Targeting Capacity for In Vivo Applications

The inherent ability of DNA based nano-materials have greater biocompatibility and biodegradability. The DNA wireframe cages and origami morphology demonstrated no pronounced toxicity and have minimum immunogenicity, meeting set basic criteria of drug delivery vehicles. Many researchers studied the stability of DNA nanostructures and they found that DNA nano-cages can enter the cells by the aid of transfection agent (lipofectamine), aggregating in the cytoplasm and maintaining the intact structure for 48 h [202]. Another study found that the DNA nanotubes with conjugated folate ligand and Cy3 fluorescent dye showed no toxicity to the live cells [203,204]. Moreover, the following study exhibited a greater targeting ability to cancer cells overexpressing the folate receptors and delivered the targeting molecule inside the cell without any transecting agents. The studies on DNA cages or origami immune stimuli hybrid nanostructures showed very low level immunogenicity of the DNA nanostructures showing the biosafety of DNA materials. The structural stability of the drug carrier in the physiological environment should be controlled carefully and not only protect its unexpected leakage on the way to tumor regions along with stimulating the drug release at the target sites. DNA tetrahedral cages showed stability in 50% non-inactivated fetal bovine serum after 4 h incubation and digested eventually within 24 h, whereas the duplex DNA in the same concentration was completely degraded within 2 h incubation. DNA origami was found to be stable in cell lysates after 12 h incubation and slowly digested by live cells within 72 h incubation [205]. These show the ability of DNA nanostructures for controlled release of therapeutic agents including siRNA, anti-body proteins. The quick and unwanted clearance of nanoparticles is of great challenge for DNA nano-carriers to stay in physiological environments due to many reasons [205]. First and foremost is the degradation of DNA via DNases (deoxyribonucleases) in serum. The other reason is that the ionic strength is quite different from the DNA assembly buffers. Hahn et al. reported that the low cationic concentration may cause the disassembly of DNA nanostructures due to enhanced electrostatic repulsion among negatively charged DNA helices. The opsonization effect via non-specific adsorption of serum proteins induces macrophages to engulf DNA nano-carriers for clearance. The DNA nanostructures have a greater ability for quick clearance via hepatic and renal clearance. There are many strategies to meet the challenges that DNA nanostructures face including designing the structures with a wireframe geometry that show greater resistance to nuclease degradation or cation depletion induced structure disassembly. There are a lot of other strategies that are available to increase the bio-stability of DNA nanostructures [205,206].

After the systemic circulation, another big issue for DNA nanocarriers is the targeted delivery to the specific tissues/organ. The specific aggregation of drug molecule on the targeted sites not only stimulate its therapeutic efficacy but also allay off-target delivery related systemic toxicity. The targeting functionality is a key factor determining the therapeutic performance of drug delivery systems. To investigate the in vivo ability of DNA nanocarriers many studies have examined their biodistributions in different animal models [207]. The pristine DNA nanostructures have accumulation in organs like kidneys, liver and lymph nodes. One of the simplest DNA structures, the DNA tetrahedrons, have been broadly researched [208]. Kim K. R. et al. reported the tendency of hepatic accumulation of Tds after intravenous intervention for liver delivery of siRNA successfully, which targets the overexpressed ApoB1 mRNA in hypercholesterolemia. ApoB1 siRNAs were encapsulated on Tds via DNA linkers incorporate from each side. About a 20–30% reduction in serum lipid levels was noticed when comparison was performed with PBS [209]. A group of researchers studied the bio-distribution of radiolabeled DNA origami nanostructures via positron emission tomography imaging of three different dimensions including triangle, rectangle and tube. All three exhibited predominant renal uptake in which triangle DNA nanostructures were applied to effectively treat the kidney injury in the mice model. The negative surface charge and size of Tds were found to be quite suitable for lymphatic drainage. Moreover, Tds can be easily taken up and have good intracellular stability, which extend their lymph node residence duration [210]. Kim et al. labeled Tds with Cy5 fluorophore to see the sentimel lymph nodes and enhanced residence time in mice xenograft models. DNA nanostructures impact on enhanced permeation and retention effect has been reported [211]. Zhang and his team reported that triangle shaped DNA origami showed optimal tumor passive targeting accumulation in comparison to rectangular and tubular origami structures. Triangle shaped origami aggregated at the tumor site and reached peak levels at 6 h and maintained high levels for 24 h after intravenous administration [212]. Kim and his colleagues developed DNA objects of different backbones for in vivo screening. Cages with backbone modifications had better tumor accumulation. The pyramid-shaped nano-cages showed greater tumor delivery efficiency. The molecular ligands used for the targeted delivery include aptamers, antibodies, functional peptides, etc. The DNA nanocarriers via different conjugation methods realize targeting functionality. The active targeting functionalized DNA nano-carriers increase the brain permeability when passive delivery fails due to the blood–brain barrier. The modified frame work of tetrahedral DNA nano-probes with peptides targeting and malignant glioma and both brain capillary endothelial cells. A study suggested that the DNA nanoprobes successfully passed via the BBB model and then entered the cytoplasm of the tumor cells [213]. Despite the prominent selectivity in vitro some ligands lose targeting capacity in vivo. One reason for the failure of the targeting ability in complex biological processes is the existence of biotransformation when adsorbed via serum proteins. This not only abolishes molecular recognition capability but also encourages clearance via phagocytes. The optimization and systemic studies need to be conducted to demonstrate the targeting ability of different ligands for in vivo delivery applications. A well-defined assays to quantify their in vivo stability need to be developed, for its potential applications in cellulo as well as in vivo. A fluorescence method can be used to evaluate the stability of DNA based nanostructures in a living organism and show that this assay shows variations between the stabilities of different DNA architectures [47,214].

For the DNA nanostructures to perform the intended functions in biological environments, they should maintain structural integrity both in vivo and in vitro. The divalent ions (below 2 mM) low concentrations can potentially inhibit its stability in biological fluids and the presence of degrading enzymes such exo-nucleases and endonucleases [215]. The DNA structures thermal annealing is usually supplemented with a greater concentration of Mg^2+^ (>10 mM). This can screen the electrostatic repulsion between the tightly packed DNA double helices, whose phosphate backbones are negatively charged. The small angle X-ray scattering experiments revealed disassembly of the 24-helix nano-rods in buffer below 2 mM Mg^2+^ [216]. The structural consistency of DNA nanostructures can be maintained for many weeks. The counter-intuitive can be explained via two effects (i) Mg+2 ions that are strongly related with the folded DNA structure after buffer exchange (ii) the buffer components such as Na_2_HPO_4_ compete for these ions and destabilize the structures. Many studies showed the fate of DNA based nanostructures in cell culture media or in in vivo experiments where limited divalent ions are present. A group of researchers tested the structural integrity of different DNA origami structures including a DNA octahedron, a 6-helix bundle and the 24-helix nanorod in an unmodified (RPMI) Roswell Park Memorial Institute tissue culture medium that consists of 0.4 mM Mg^2+^ [215]. The structures showed various levels of stability scaling depending on their packing density. Adding the medium with 6 mM Mg^2+^ preserved the structural integrity of all structures while maintaining cellular viability. This study results showed that the 6-helix bundle tubes made of 42-base single stranded DNA tiles are stable in as low as 2 mM Mg^2+^ concentration in PBS buffer and also in Dulbeccos’ Modified Eagle Medium for 8 h. A group of researchers studied the stability of DNA nanostructures and reported that the DNA polyhedral meshes that consist of loosely packed single helices, stay intact in biologically relevant buffers. Single stranded DNA nanostructures also demonstrated an increased stability against agents like urea and guanidinium chloride compared to double stranded DNA [217]. The structural integrity of DNA nanostructures is relevant to the temperature, the ion concentrations in the surrounding media and the DNA building blocks length and packing density of DNA helices with tightly packed DNA structures at 37 °C. DNA stability of nanostructures against catalytic enzymes can be tested by mixing DNA structures with isolated nucleases, cell lysate or blood serum [218]. Mei et al. reported DNA origami structures by TEM and atomic force microscopy after exposing them to the cellular lysate. The structures were intact after 12 h of incubation at room temperature. However, single or double stranded DNA nano-structures in contrast degraded within 1 h of incubation [219]. Castro et al. showed the stability of DNA origami bundles against several nucleases including DNAse I, T7 endonuclease I, T7 exonuclease and MseI restriction enzyme. The authors reported that DNAse I degrades 2 ng of DNA origami structures in 60 min at 37 °C, while complete degradation of 65 ng of plasmid DNA takes only 5 min. The enzymes did not cause any structural impairment to the origami structures [217]. The plasmid DNA nanostructures have packed DNA double helices that inhibit access to the nucleases. One group of researchers reported that DNA tetrahedrons are stable against the enzyme DdeI, when the recognition site is close to the vertexes. All the reported studies showed that DNA nanostructures show increased resistance against nucleases by binding or recognition of the site. The stability of DNA nanostructures can be enhanced by covalently linking strands using chemical groups or by loading the entire structures with a lipid bilayer. A group of researchers photo-crosslinked the thymine bases and 8-MOP (8-methoxypsoralen) and enhanced the thermal stability of DNA origami tiles upto 60 °C. The click reaction method was used to insert the covalent bonds (copper (I)-catalyzed) alkyne-azide cycloaddition [220]. The formamide-induced degradation show copper-free click reactions on 3-alkyne and 5-azide-oligonucleotides. A group of researchers reported that the 6-helix tubes consisting of 24 catenanes maintain the integrity at extremely harsh conditions such as temperatures of 95 °C, presence of exonuclease I and lack of any ions in the buffer. One more method used to enhance the stability of DNA structures is the formation of disulfide bonds among adjacent DNA strands. The DNA origami structures were digested within 24 h when the cell medium was added with more than 5% FBS (fetal bovine serum). The authors reported that to prevent the nuclease activity during in vitro experiments is to heat inactivate the nucleases at 75 °C and to add actin proteins to inhibit the nuclease activity while not affecting the cell viability and growth. A study conducted reported DNA octahedrons loaded with PEGylated lipid bilayers via using lipid modified oligonucleotides onthe external part of the origami structures as a nucleation site for membrane formation [221]. The reported results demonstrated that encapsulated DNA octahedrons showed 17 times greater half-life in comparison to DNA oligonucleotides and octahedrons alone. The results showed that DNA nanostructures of complex shape can be modified and show withstand conditions of low divalent ions inthe presence of nucleases and enhanced temperature. The following mentioned stability strategies increase the DNA nanostructures in blood serum and cell media for the employment of these structures as an efficient carrier systems with efficient functionality in biological studies [222].

## 9. Conclusions, Challenges and Future Aspects

In conclusion, smart stimuli-responsive DNA nanostructures have a wide range of possibilities to construct into various shapes and sizes. DNA was used as a template to build up a structure either tile-based, origami-based or supramolecular. Synthesis strategies of the smart DNA nanostructures are classified based on various types of DNA nanocarriers like self-assembled DNA hydrogels, DNA origami-based nanorobots, DNA-linked metallic nanoparticles, i-motif DNA supramolecular nanostructure, DNAzyme based nanostructures, DNA nanowires, and DNA-liposomes. Additionally, stimuli-responsive bonds, proteins, peptides, chemical linkers, oligonucleotides, aptamers, etc. were incorporated in these structures during fabrication to endowed DNA nanostructures with specific features that specifically target the cancer cells and released the payload. Multiple mechanisms of cellular uptake have been described for different types of DNA-based carriers, mostly caveolin-mediated endocytosis, however, it is still debatable. The recent applications of nanotechnology have made profound impacts on the development of nano-medicines. The nanocarriers bring a novel strategy for the delivery of bioactive compounds to tumors, so that the precise drug release will be better carried out which is beneficial to cancer therapy and diagnosis. The DNA nanostructure, stimuli sensitive and hybrid nano-carrier provide high specificity and multiple functions in drug delivery. Besides the rational design of DNA nanostructure and stimuli responsive nanocarriers considered their biological manner in tumor cells to enhance the efficacy and reduce the adverse effects to normal tissues. DNA based nanostructure and stimuli sensitive nanostructure can widely applied for tumor therapy and for diagnosis. However, maintaining the stimuli sensitivity on large scale would be a potential challenge. Furthermore, with wide spectrum studies, only a few formulations have entered clinical translation, which need future extensive research work. Some challenges that are required to tackle before DNA nanostructure systems are used for drug delivery in humans. For example, we should address the pharmacokinetics of DNA nanostructure vehicles including, distribution, metabolism, excretion. The physical and chemical properties including surface charge, geometry, base combinations should be closely observed as these can affect the pharmacokinetics, bioavailability of DNA nanostructures. These studies are still required to implement DNA nanostructure clinically.

## Figures and Tables

**Figure 1 cancers-13-03396-f001:**
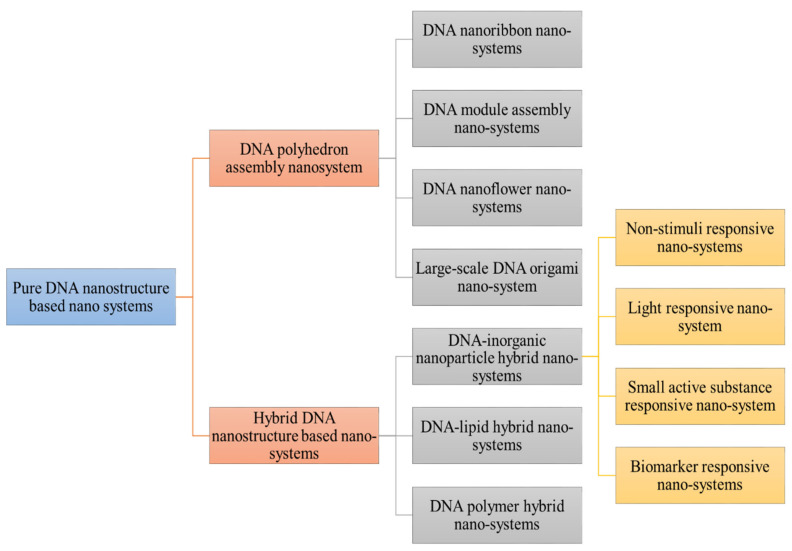
Demonstrated different classification of DNA nanostructure. Pure DNA nanostructure is divided into DNA polyhedron assembly nanosystem (DNA nanoribbon, DNA module assembly nanosystem, DNA nanoflower system) and hybrid DNA nanosystem (DNA-inorganic nanoparticle hybrid nanosystems (non-stimuli responsive, small active responsive, biomarker responsive) DNA lipid hybrid and DNA polymer hybrid nanocargo.

**Figure 2 cancers-13-03396-f002:**
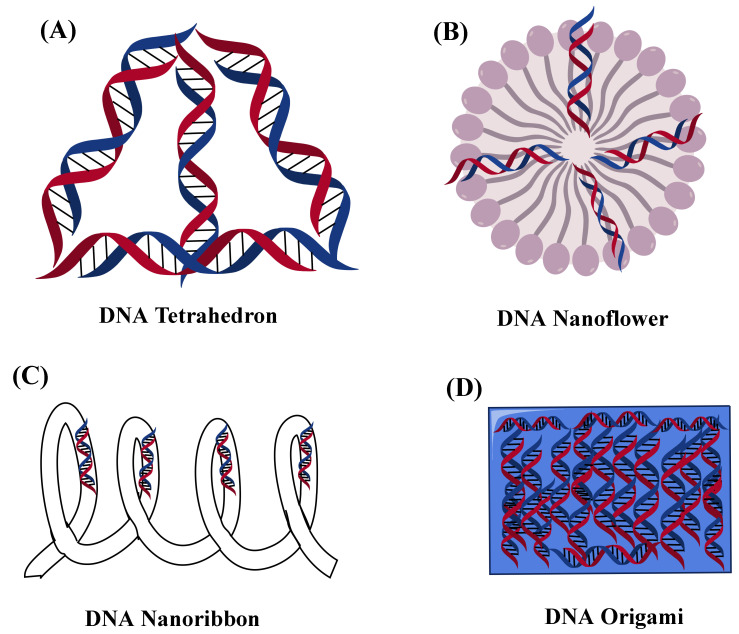
Showed the nanostructure representation of different types of DNA (**A**) DNA tetrahedron, (**B**) DNA nanoflower, (**C**) DNA nanoribbon, and (**D**) DNA Origami.

**Figure 3 cancers-13-03396-f003:**
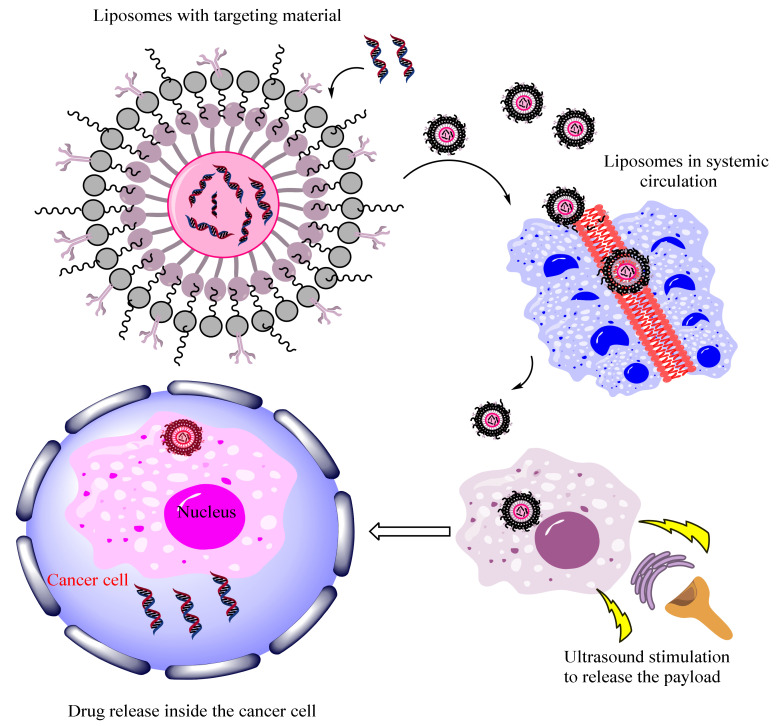
Demonstrated the application of ultrasound responsive liposome-carrier system for cancer targeting.

**Figure 4 cancers-13-03396-f004:**
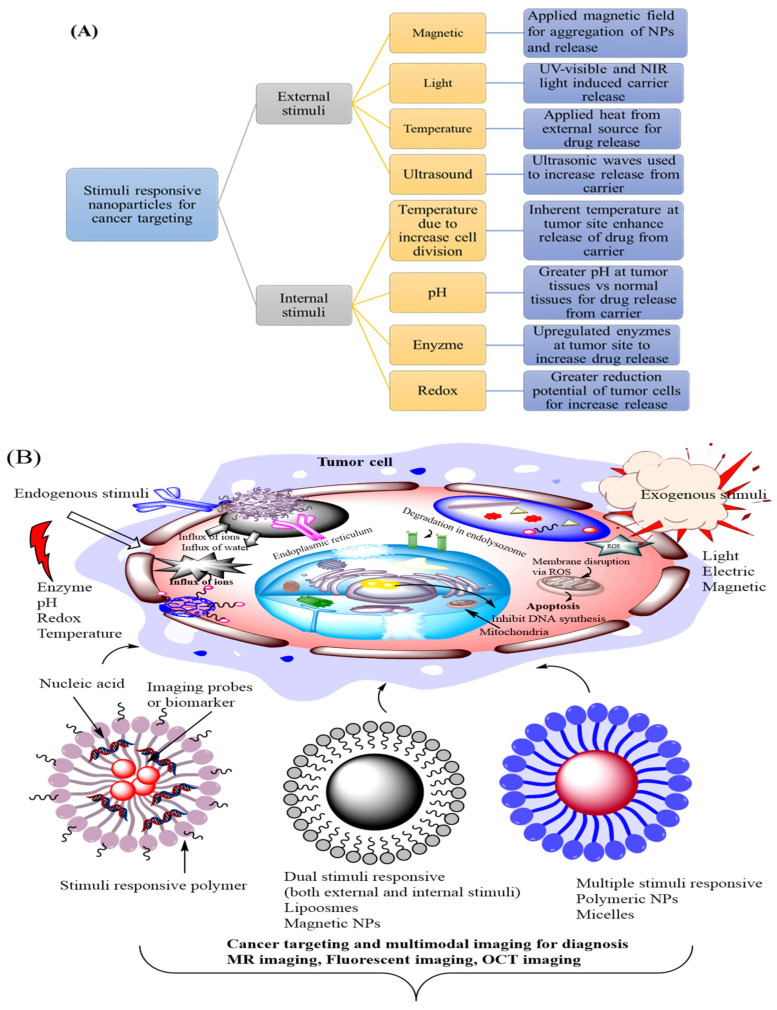
(**A**) Schematic representation of different types of exogenous and endogenous stimuli (**B**) Showed different types of stimuli responsive nanoparticle for cancer targeting.

**Figure 5 cancers-13-03396-f005:**
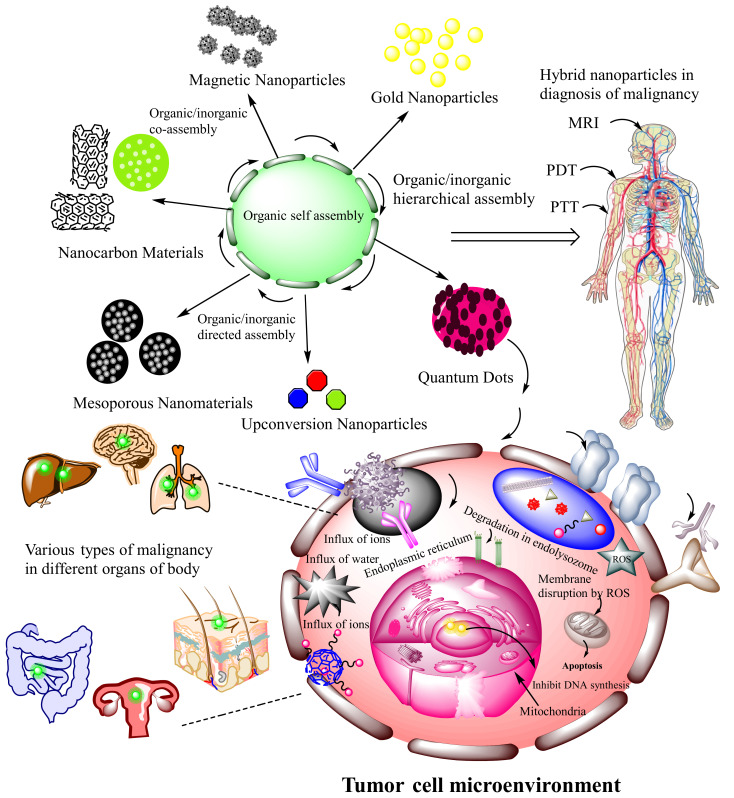
Showed types of hybrid nanoparticles for tumor targeting and diagnosis.

**Figure 6 cancers-13-03396-f006:**
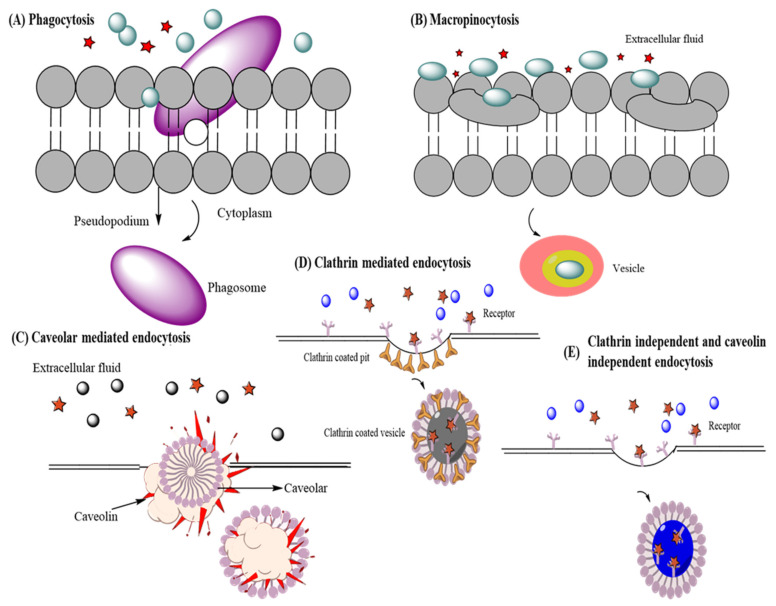
Mechanism and different pathways of internalization of DNA conjugate nanoparticles. (**A**) Phagocytosis; (**B**) Macropinocytosis; (**C**) Caveolar mediated endocytosis; (**D**) Clathrin mediated endocytosis; and (**E**) Clathrin independent and caveolin independent endocytosis.

**Table 1 cancers-13-03396-t001:** Prerequisite of DNA nanostructures to fulfill the mean particle diameter for specific organ targeting.

Targeting Site	Mean Particle Diameter	Surface Characteristics	Ref
Bone	Undefined	Substances like aspartic acid, alendronate can adhere to bone and can be used for bone targeting.	[59]
Liver	Less than 100 nm to cross the liver fenestrae and target the hepatocytes. Greater than 100 nm uptake by Kupffer cells.	No define surface property needed	[59,60]
Lung	Particles larger than 200 nm are trapped into lung capillaries	Cationic surface charge	[61]
Brain	5–100 nm: nanoparticles uptake efficiency decreases with size	Hydrophobic moieties and neutral charge enhance the brain uptake	[59,62]
Lymph nodes	1–40 nm: intra-tracheal administration80 nm: Subcutaneous application	Non-pegylated, Non-cationic and sugar based particles.	[59,60]

**Table 2 cancers-13-03396-t002:** Exogenous and endogenous stimuli and nanocarriers system for gene and drug delivery at tumor site.

Exogenous and Endogenous Stimuli and Delivery System	Encapsulated Moiety	Application	Advantages	Limitation	Ref
NIR light ✓Carrier free nanosystem	DOX	Ablation of tumor via photo thermal chemotherapy	Easily tuned, Deep penetration, greater precision, no damaging, minimally invasive	Ionization radiation, Expensive equipment	[65]
✓Mesoporous silica nanoparticles	DOX and Camptothecin	Photodynamic and Chemotherapy
Ultrasound nanoparticles✓Microbubble	DOX	Targeted drug delivery to the tumor site	Low cost, greater patient compatability, no ionizing radiations	Difficult to remove the remote and moving targets	[66,67]
✓nanoparticle aggregate	siRNA	Image modulated therapy
Magnetic field ✓Solid lipid nanoparticles✓Lipid coated superparamagnetic nanoparticles	Paclitaxel, Curcumin, Camptothecin	Targeted delivery against tumor imaging and therapy, targeted delivery by magnetic hyperthermia	No ionizing radiation, deep penetration, imaging opportunity, energy modulation with an atomic force microscopy (AFM)	Expensive, limited to the surface tumors, increased cytotoxicity, accumulation can lead to emboli formation	[65,68]
Temperature ✓Selfheable hydrogel✓Nanogel	DOX and curcumin	Targeted drug release	High mobility of matrix, High precision, inexpensive	Limited tissue penetration	[37]
pH ✓Polymeric nanoparticles✓Liposomes	Plasmid DNA	Cytoplasmic delivery	Cationic polymer induces membrane fusion at endosomal pH, Improved anti-cancer property in murine tumor model, Increased gene transfection to hepatocytes	Heterogeneity and diversity of cancer cell can limit the targeted delivery	[69]
Redox sensitive ✓Liposomes✓Thiopolycation PESC (PHEA-EDA-SH-CPTA)	Plasmid DNA	Targeted delivery	Thioplexes release DNA in reductive environment	Heterogenicity of cancer cells and accumulation of nanoparticles may cause toxicity	[70,71]
Temperature sensitive ✓Poloxamer liposomes	Lucifer yellow Iodoacetamide	More than 90% release was achieved at 42 °C at targeting site	Showed several fold increase in targeting moiety in tumor bearing mice	Heterogenicity of cancer cells, Toxicity of nanoparticles inside the vital organs	[72,73]

**Table 3 cancers-13-03396-t003:** Summary of DNA synthesis method of smart DNA based nanostructures that are responsive to different stimuli.

No.	Type of DNA Nanostructure	Synthesis Method	Stimuli-Responsive Unit	Targeting Unit	Stimuli	Drug/Encapsulant	Outcome	Reference
1.	Self-assembled DNA hydrogel	Self-assembly of DNA sticky ends through a linker (hydrogel)	Di-sulfide bond	Aptamer/antisense oligonucleotide	Glutathione (GSH) enzyme	--	Gene regulation in (human lung adenocarcinoma (A549) cell lines	[151]
2.	DNA nanorobot	DNA origami-based synthesis	Aptamer-nucleolin interaction	Aptamer	Endogenous tumor microenvironment (nucleolin) factors trigger DNA nanorobot untangling	Thrombin	Tumor cell inhibition through targeted delivery of thrombin	[155]
3.	DNA capped M-SiO_2_/Fe_3_O_4_/Au nanoparticles	Covalent linkage between dsDNA and surface modified M-SiO_2_/Fe_3_O_4_/Au nanoparticles	Aminopropyltriethoxysilane, Fe_3_O_4_, Au	--	Photothermal NIR/Magnetic	DOX		[157]
4.	Aptamer-i-motif DNA-Au nanoconjugates	DNA supramolecular i-motif-Au thiol linkage (salt aging process)	i-motif, Au, Aptamer-nucleolin interaction	Aptamer	Tumor pH + photothermal NIR	DOX	Targeted tumor cell ablation and pH mediated drug release	[159]
5.	i-motif DNA-Au nanoparticles	Modified oligodeoxynucleotides i-motif linkage to citrate capped Au nanoparticles via thiol group (salt aging process)	i-motifs	BCL-2 antisense oligodeoxynucleotides	Tumor pH	DOX	pH mediated drug release at the tumor, cancer cells apoptosis	[160]
6.	DNAzyme modified M-SiO_2_ coated Au nanorods	EDC/NHS chemistry	4,4′-azobis(4-cyanovaleric acid) (AC)	Survivin-DNAzyme	photothermal NIR	DOX	Improve sensitization of triple resistant breast cancer cells towards phototherapy	[162]
7.	Self-assembled DNA nanowires	DNA oligonucleotide hybridization reaction	chlorine e6 (ce6)	--	Photodynamic (monochromatic light)	DOX	Chemo-photodynamic therapy ablates cancer cells	[96]
8.	N-APT-liposome	Cholesterol led immobilization of DNA aptamer on lipids	Aptamer-nucleolin interaction	Cholesterol tagged N-Apt	Tumor endogenous nucleolin	Cisplatin or dye	Tumor-specific drug delivery to inhibit proliferation	[163]

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
