# Peer review of "DNA Based and Stimuli-Responsive Smart Nanocarrier for Diagnosis and Treatment of Cancer: Applications and Challenges"

_cancers, 2021, doi:10.3390/cancers13143396_

Round 1

Reviewer 1 Report

The Authors sufficiently revised the manuscript, improving the quality and adding the required information. 

Author Response

Comments: The Authors sufficiently revised the manuscript, improving the quality and adding the required information. 

Response: We would like to thanks reviewer for their time and previous positive comments and fully satisfy with our response made by authors.

Reviewer 2 Report

I still find a series of sentences, especially in the section 2 and 3 that in my opinion are difficult to understand. Here just some examples:

Line 133. When CTC-bound DNA nanodevices flow through the superficial capillary.

  1. …and also enhanced the drug loading capacity and govern synergise therapy that enhanced their destructive nature

Line 164 The development of functionalized DNA nanoribbon is the most important function of DNA nanoribbon in anti-cancer drug delivery.

Line 178 Through base pairing hybridization layer by layer self-assembled functional branched of DNA is called DNA dendrimers

Line 184 nanohydorgels: change

Line 186 Different researchers have applied different method for the development of DNA nanohydrogels for trageting Dox delivery from by using building blocks and via using liquid crystallization without base pairing hybridization

Line 192 DNA nanoflower system in comparison to its self-assembled from long DNA strands via rolling circle replication along with liquid crystallization and dense packaging

Line 198 Furthermore, the researcher modify the nanoflowers to encapsulate 2 kinds of multigene therapy and DNAzymes

Line 215 DNA further classified into hybrid DNA nanostructured based nanosystems that is subdivided into DNA-inorganic nanoparticle hybrid (non-stimuli responsive, light responsive, small molecule, DNA lipid hybrid, DNA polymer hybrid nanosystems, small 216 active substance responsive nanosystem)

Line 218 DNA-inorganic nanparticles hybrid nanosystems including non-stimuli responsive system. DNA-inorganic nanoparticle based nanosystems designed for the better cancer treatment.

Line 222 Nanoflower connected these nanoparticles with inorganic nanoparticles for spherical structure and increased encapsulated amount of drug for good cancer targeting, according to the researchers

Line 246 Another type of DNA nanocargo include polymer hybrid nanosystems this system have strong encapsulation affinity and they can with load protect the drug against premature degradation.

Line 249 Willner’s et al, developed DNA polymer hybrid system for controlled release based on poly function as the core and multiple layers as shell.

Line 306 In another research, liposomes encapsulated with docetaxel and NH4HCO3 to generate gas in tumors for dual ligand targeted therapy and ultrasound imaging.

Line 307 One study claimed multimodal ultrasound imaging and molecular biosensors application of  nanodroplets gas vesicles by using genetically encoded gas nanostructure from microorganism

Line 603 Oligonucleotides such as siRNA and microRNA, which are powerful active agents that have been used to deliver oligonucleotides into tumors.

Please, check references:

Line 166 Liang et al: reference?

[35] is Roh, not Wang

A revision of English must be performed: some mistakes are still present. Some are mispelling (see above) but in some cases there are problems with tenses, or subject-verb concordances. In some sentences the verb is missing

Reviewer 3 Report

The manuscript “DNA based and stimuli-responsive smart nanocarrier for diagnosis and treatment of cancer: applications and challenges” by Sabir et al. summarized various types of DNA nanostructures and stimuli responsive nanocarrier systems for diagnosis and treatment of cancer. I would suggest authors may take at least a minor revision before publication. Here are the comments and suggestions:

  1. Line 288, the size of microbubbles is incorrect. It should be in microns.
  2. Line 30~, the work in the International Journal of Nanomedicine, 7, 941-951, 2012 is suggested to add in the system of nanocarrier emulsion.
  3. In fig. 3, there are two tails for each lipid on liposomes. Is of the caption of fig.3 “ nano-carrier system” or “liposome system”

Author Response

Reviewer# 3

The manuscript “DNA based and stimuli-responsive smart nanocarrier for diagnosis and treatment of cancer: applications and challenges” by Sabir et al. summarized various types of DNA nanostructures and stimuli responsive nanocarrier systems for diagnosis and treatment of cancer. I would suggest authors may take at least a minor revision before publication. Here are the comments and suggestions:

Response: We would like to thanks reviewer for their time and positive comments on the importance of our work. We also thank them for their constructive remarks, which we have acted upon to help further improve the manuscript. Please find below our point-by-point response to the reviewers’ comments and the corresponding modifications we have made to the manuscript, which we hope will make it acceptable for publication in Cancers.

1. Line 288, the size of microbubbles is incorrect. It should be in microns.

Response: Correction done. We have carefully revised the manuscript to ensure that the text is optimally phrased and free from typographical and grammatical errors.

2. Line 30~, the work in the International Journal of Nanomedicine, 7, 941-951, 2012 is suggested to add in the system of nanocarrier emulsion.

Response: Correction done. Thanks we added this article please check the improve version.

3. In fig. 3, there are two tails for each lipid on liposomes. Is of the caption of fig.3 “ nano-carrier system” or “liposome system”

Response: Correction done. Thanks a lot for pointing out this error its liposome system we made corrections in the revised manuscript.

Round 2

Reviewer 2 Report

Most of previous criticisms have been addressed. I find the present form more clear.